# OBSERVE-THEN-THINK:
# LEARNING TO ELICIT MULTIMODAL UNDERSTANDING BY DECOUPLING PERCEPTION AND REASONING

## ABSTRACT

Vision-language models (VLMs) have achieved impressive results in fusing visual and textual inputs, yet they often stumble on tasks demanding complex, multimodal reasoning. This imbalance arises from the inherent separation between perception—accurately interpreting sensory data, and reasoning—conducting multi-step, symbolic inference. To bridge this gap, we introduce a novel framework for multimodal reasoning, *OTT (Observe-Then-Think)*, which includes a two-stage post-training process: supervised fine-tuning (SFT) followed by reinforcement learning (RL). During SFT, the model learns to decouple perceptual understanding from logical inference, mastering structured output formats and ensuring logical consistency. In the RL stage, our Perception-Guided Consistency Optimization (PGCO) algorithm, inspired by human cognition, enhances visual understanding through perception rewards and employs consistency rewards to align perception and reasoning steps, ensuring the final answer accuracy, eliminating logical contradictions without external tool support. Extensive evaluations across six challenging benchmarks demonstrate that our method consistently outperforms state-of-the-art baselines by an average of 3.8% over the baseline models, delivering both stronger perceptual grounding and more reliable multimodal reasoning of VLMs.

## 1 INTRODUCTION

Multimodal reasoning with vision-language models (VLMs) aims to bridge perception and abstract, multi-step inference, enabling interpretation of real-world artifacts such as scientific figures, charts, and visual narratives. Recent advances in Gemini 2.5 (Comanici et al., 2025) and Qwen2.5-VL (Bai et al., 2025) are making integrated multimodal reasoning increasingly practical for scientific analysis, physical interaction, and counterfactual reasoning. Yet VLMs still struggle with *complex, multi-step problems* (e.g., multimodal mathematical reasoning): *perception* processes sensory inputs, whereas *reasoning* requires compositional, symbolic inference, and prevailing architectures often favor perceptual alignment over reasoning depth. Progress is further constrained by the scarcity of high-quality multimodal reasoning datasets for real-world analysis and interaction, motivating post-training of pretrained VLMs on targeted reasoning corpora to tackle complex downstream tasks.

Reinforcement learning (RL) post-training has markedly improved model reasoning, with Group Relative Policy Optimization (GRPO) (Shao et al., 2024) particularly effective. Unlike supervised imitation of step-by-step traces, GRPO uses rule- or model-based rewards to let models discover high-quality reasoning paths, optimizing outcomes while reducing annotation cost. RL already shows strong gains in text-based reasoning, code generation, and formal logic, and early efforts suggest similar benefits for the *reasoning* capabilities of VLMs. However, VLMs must tightly couple both *perception* and *reasoning*, and limited perception remains a key *bottleneck* (Lu et al., 2024; Liu et al., 2025a). This raises our central question: *Can RL not only sharpen the reasoning power of VLMs but also elicit stronger perceptual abilities, enabling more comprehensive multimodal reasoning?*

To close the perception-reasoning gap, we propose jointly incentivizing both capacities, guided by neurobiology. In the brain, visual processing originates in the *occipital cortex* and proceeds along ventral ("what") and dorsal ("where/how") streams for object identity and visuomotor transformation, while abstract reasoning and cognitive control arise in *prefrontal* circuits that maintain goals and

Figure 1: The structural "observe-then-think" process for multimodal understanding and reasoning. The model first observes the image to extract relevant visual information, then reasons over the question based on the observation, and finally generates a grounded answer.

modulate downstream processing. Attention prioritizes relevant features, working memory sustains them for deliberation, and predictive-coding dynamics separate bottom-up evidence from top-down hypotheses. Inspired by this division of labor, we propose *OTT (Observe-then-Think)* for multimodal reasoning (Fig. 2): the model first *observes*—producing a structured, perceptual sketch—then *thinks* through step-by-step thoughts conditioned on that sketch, and finally *answers* in a precise format.

Decoupling perception from reasoning brings three concrete benefits: (1) *clearer credit assignment:* stage-specific objectives/rewards isolate whether failures are perceptual or logical; (2) *verifiability and interpretability:* the perceptual sketch explicitly exposes intermediate thoughts about visual information that can be further checked or revised; and (3) *robustness and generalization:* reasoning operates on purified evidence, reducing shortcuts to language priors and potentially improving out-of-distribution transfer. Together, these advantages allow OTT to better mirror cortical separation of evidence accumulation and top-down control, enabling stronger multimodal reasoning in VLMs.

To realize the *observe-then-think* paradigm, we adopt a two-stage training framework. First, we perform supervised fine-tuning (SFT) on **OTT-20k**, a curated set of 20k standardized examples distilled from Mulberry-260k, to teach strict structural separation of outputs into *observe*, *think*, and *answer*, reinforcing a disciplined OTT workflow. Second, we apply **perception-guided consistency optimization** (PGCO), a GRPO-based RL procedure with four complementary rewards: (i) answer accuracy, (ii) format compliance, (iii) perceptual fidelity of the *observe* sketch to the image, and (iv) consistency between *observe* and *think*. Together, structure-aware SFT and perception-aware RL yield VLMs that generate observe-then-think trajectories and excel on multimodal reasoning tasks.

Our experiments demonstrate that the two-stage OTT framework delivers consistent, state-of-the-art gains across a wide range of visual reasoning tasks. We evaluate on six diverse benchmarks, *i.e.*, MathVista, WeMath, BLINK, V*Bench, VisuLogic, and MME, that collectively span diagrams, plots, equations, tables, and OCR-heavy inputs, and challenge models with symbolic, numerical, logical, and commonsense reasoning. To verify scalability, we apply our method to Qwen2-VL-7B-Instruct, observing clear improvements over strong baselines regardless of task modality. All evaluations are conducted under the unified VLMEvalKit framework, with Qwen2.5-VL-32B-Instruct serving as the judge for mathematical problems. These results confirm that our approach not only enhances interpretability through structured "observe-then-think" but also achieves top-tier performance on fine-grained, multimodal benchmarks.

In summary, our main contributions are three-fold as follows:

- **Introduction of the OTT Paradigm.** We introduce the Observe-then-Think (OTT) paradigm, a novel multimodal reasoning framework that explicitly separates perception from reasoning, enabling VLMs to first observe and interpret visual inputs before engaging in structured, step-by-step reasoning. To support this approach, we present OTT-20k, a meticulously curated supervised fine-tuning dataset containing 20k examples to reinforce this structured reasoning workflow (Sec. 3.2).

- **Perception-Guided Consistency Optimization (PGCO).** We propose the Perception-Guided Consistency Optimization algorithm, a GRPO-based reinforcement learning strategy that integrates perception rewards and consistency rewards to tightly couple visual perception with accurate reasoning, improving the consistency and transparency of multimodal reasoning traces (Sec. 3.3).

- **Comprehensive Empirical Validation.** Our approach demonstrates consistent, state-of-the-art performance across six challenging multimodal reasoning benchmarks, including MathVista,

WeMath, BLINK, V*Bench, VisuLogic, and MME. These evaluations confirm that the OTT framework not only enhances interpretability through structured "observe-then-think" reasoning but also achieves superior performance across diverse task modalities (Sec. 4).

## 2 PRELIMINARIES

**Post-training** is the process of refining a foundation model, which has already been extensively pre-trained on large and diverse datasets, to improve its performance, adapt it to specific tasks, and align it with human values. A post-training dataset $\mathcal{D}$ consists of question-answer pairs $(q, a)$. Given a question $q$, a language model $f_{\boldsymbol{\theta}}(\cdot)$, parameterized by $\boldsymbol{\theta}$, generates an output $o$ as $o \sim f_{\boldsymbol{\theta}}(\cdot \mid q)$, where the output $o$ includes both the model's step-by-step reasoning process and the predicted answer. In this context, models are often categorized by the frequency of their parameter updates: the policy model $f_{\boldsymbol{\theta}}$, which is the most recently updated version; the old policy model $f_{\text{old}}$, an earlier version of the policy model; and the reference model $f_{\text{ref}}$, the oldest and typically used for comparison.

**Group Relative Policy Optimization (GRPO)** (Shao et al., 2024) is a PPO-style algorithm for language models that removes the learned value function and instead uses a group-relative baseline. For each question $q$, draw $G$ candidate outputs $\{o_i\}_{i=1}^{G}$ from the old policy $f_{\text{old}}$. Score each candidate with a deterministic reward $r_i$ (e.g., accuracy), then compute a group baseline $\bar{r}$ from these $G$ rewards and form advantages $\hat{A}_i$ as standardized deviations from $\bar{r}$. The current policy $f_\theta$ is updated to increase token log-likelihood on candidates with positive $\hat{A}_i$ and decrease it for negative ones, using a PPO-style clipped likelihood ratio $\rho_{i,t}$ for stability together with a KL penalty $\text{KL}(f_\theta \| f_{\text{old}})$ to remain close to the reference policy. A detailed introduction of the related papers in Appendix E.

## 3 METHOD

### 3.1 OBSERVE-THEN-THINK PARADIGM

We propose an *observe-then-think* paradigm that explicitly separates perception from reasoning. Given a multimodal input consisting of a question $q$ and an associated visual context $v$, the policy model $f_{\boldsymbol{\theta}}$ first generates an *observation sequence*: $o_{\text{obs}} \sim f_{\boldsymbol{\theta}}(\cdot \mid q, v)$, which captures the salient visual details required for subsequent reasoning. The model then engages in a structured chain-of-thought process, producing a *reasoning sequence*: $o_{\text{think}} \sim f_{\boldsymbol{\theta}}(\cdot \mid q, v, o_{\text{obs}})$, where each step corresponds to a logically coherent intermediate inference. Finally, conditioned on both the observation and reasoning traces, the model outputs a final answer: $o_{\text{ans}} \sim f_{\boldsymbol{\theta}}(\cdot \mid q, v, o_{\text{obs}}, o_{\text{think}})$. To enforce this disciplined separation, the outputs are required to follow a structured format with XML-style tags:

$$f_{\boldsymbol{\theta}}(\cdot \mid q, v) \mapsto \langle\text{observe}\rangle \, o_{\text{obs}} \, \langle/\text{observe}\rangle, \quad \langle\text{think}\rangle \, o_{\text{think}} \, \langle/\text{think}\rangle, \quad \langle\text{answer}\rangle \, o_{\text{ans}} \, \langle/\text{answer}\rangle,$$
$$\text{where } o_{\text{obs}} \sim f_{\boldsymbol{\theta}}(\cdot \mid q, v), \quad o_{\text{think}} \sim f_{\boldsymbol{\theta}}(\cdot \mid q, v, o_{\text{obs}}), \quad o_{\text{ans}} \sim f_{\boldsymbol{\theta}}(\cdot \mid q, v, o_{\text{obs}}, o_{\text{think}}).$$

For example, given an image and a question, "which item sold the most number of units summed across all the stores?", we expect the model to output in the following "observe-then-think" manner:

> **<observe>** The image is a horizontal bar chart showing sales statistics for items in different stores. It displays sales for two types of stores named "theory" and "impact". The x-axis represents units sold, with a range from 0 to 10. The items listed on the y-axis are "shot", "smell", "editor", "space", "career", "bulk", and "rush". Two bars per item illustrate sales in both stores, with "theory" in black and "impact" in gray. **</observe>**
> **<think >** The question asks to find which item sold the most units when summed across all stores, requiring the addition of units sold in both "theory" and "impact" stores for each item, then identifying the maximum. Let's think step by step.
> Step 1: Identify the units sold for each item in the "theory" store from the bar chart ...
> Step 2: Identify the units sold for each item in the "impact" store from the bar chart ... **</think >**
> **<answer >** career **</answer >**

This paradigm enforces a strict separation between observation, reasoning, and answer, thereby enhancing interpretability, transparency, and internal consistency—properties essential for trustworthy post-training and evaluation. We implement it through a two-stage framework: first, supervised

Figure 2: Overview of the dataset construction pipeline, including data collection, format refinement, and formation of SFT and RL datasets.

fine-tuning (SFT) to instill adherence to structured reasoning formats, and second, reinforcement learning (RL) to further strengthen complex multi-modal reasoning, as detailed in following sections.

## 3.2 SFT STAGE: TEACHING THE MODEL TO OUTPUT STRUCTURALLY

**Curating the OTT-20k dataset for SFT.** To train the model to effectively follow the *observe-then-think* (OTT) paradigm described in Section 3.2, we curate the OTT-20k dataset. OTT-20k consists of $20,000$ carefully curated samples, striking a balance between data diversity and computational efficiency. This dataset captures a broad spectrum of reasoning patterns while maintaining manageable training costs. Our data collection process builds on prior methodologies (Muennighoff et al., 2025; Yao et al., 2024a) and proceeds in two steps: preprocessing and extraction.

**Step 1: Data Preprocessing.** We begin from the Mulberry-260k dataset (Yao et al., 2024b), which spans 31 diverse benchmarks across natural image understanding, scientific figure interpretation, chart and table comprehension, document and textbook reasoning, geographical and spatial reasoning, and medical VQA. Each sample includes an input image $v$, a user query $q$, and a model-generated response $o$. Formally, a raw sample can be represented as $(q, v, o) \in \mathcal{D}_{\text{raw}}$ with $|\mathcal{D}_{\text{raw}}| \approx 260k$. To obtain a standardized pool for downstream processing, we perform three key operations:

1. *Standardizing Response Format.* Each response is transformed into a structured form that separates observation, reasoning, and answer:

$$o \mapsto \langle\text{observe}\rangle\, o_{\text{obs}}\, \langle/\text{observe}\rangle,\ \langle\text{think}\rangle\, o_{\text{think}}\, \langle/\text{think}\rangle,\ \langle\text{answer}\rangle\, o_{\text{ans}}\, \langle/\text{answer}\rangle. \quad (1)$$

2. *Atomic Answer Filtering.* We retain only those samples where the final answer $o_{\text{ans}}$ is *atomic*, i.e.,

$$o_{\text{ans}} \in \mathcal{V}_{\text{atomic}}, \mathcal{V}_{\text{atomic}} = \{x \in \text{closed-ended responses} \mid x \text{ is non-composite}\}. \quad (2)$$

Non-composite does not contain complex mixtures of words, digits, and punctuation.

3. *Deduplication and Quality Enhancement.* For each unique query $q$, we retain only the sample with the longest response, reducing redundancy and ensuring richer reasoning traces.

After preprocessing, we obtain a refined pool of $|\mathcal{D}_{\text{pool}}| \approx 170k$ samples for SFT.

**Step 2: Sample Extraction.** From $\mathcal{D}_{\text{pool}}$, we extract a balanced subset of $20k$ samples in SFT stages:

1. *Uniform Sampling Across Datasets.* We allocate the 20k target uniformly across the 31 datasets, with $n_d = \lfloor 20000/31 \rfloor \approx 645$ samples each; if a dataset has fewer, all its samples are kept.

2. *Supplementary Sampling.* If the total sample count is less than $20k$ after uniform allocation, the remaining quota is filled by random sampling from unused samples in $\mathcal{D}_{\text{pool}}$.

The resulting dataset, $\mathcal{D}_{\text{OTT-20k}} \subset \mathcal{D}_{\text{pool}}$, ensures both diversity and balance: $|\mathcal{D}_{\text{OTT-20k}}| = 20,000$.

**SFT Training.** Finally, we use $\mathcal{D}_{\text{OTT-20k}}$ to post-train a model with the standard SFT objective:

$$\mathcal{J}_{\text{SFT}}(f_{\boldsymbol{\theta}}) \triangleq \mathbb{E}_{(q,v,o)\sim\mathcal{D}_{\text{OTT-20k}}}\Big[\log f_{\boldsymbol{\theta}}(o \mid q, v)\Big], \quad (3)$$

where $o$ is the structured output containing observation, reasoning, and final answer.

## 3.3 RL STAGE: INCENTIVIZING THE MULTIMODAL REASONING CAPABILITIES

After the SFT stage, the subsequent RL phase leverages our perception-guided consistency optimization (PGCO) algorithm. Inspired by human cognitive processes, PGCO introduces perception-specific

Figure 3: Overview of the proposed method framework, consisting of two stages: SFT and RL. In the SFT stage, the model is taught to follow an "Observe-Then-Think" reasoning format using 20K supervised examples. In the RL stage, the model is further optimized with 20K reinforcement learning samples under a reward framework that includes format, accuracy, and perception rewards.

rewards to enhance visual precision, while format rewards ensure that each reasoning step aligns coherently with the final answer. The optimization objective of PGCO is as follows:

$$\mathcal{J}_{\text{PGCO}}(f_{\boldsymbol{\theta}}) \triangleq \mathbb{E}_{(q,a)\sim\mathcal{D},\{o_i\}_{i=1}^{G}\sim f_{\text{old}}(\cdot|q)} \frac{1}{G} \sum_{i=1}^{G} \frac{1}{|o_i|} \sum_{t=1}^{|o_i|} \tag{4}$$

$$\left[ \min\left( \rho_{i,t}\hat{A}_{i,t}, \text{clip}\left(\rho_{i,t}, 1-\epsilon, 1+\epsilon\right)\hat{A}_{i,t} \right) - \beta\mathbb{D}_{\text{KL}}\left[f_{\boldsymbol{\theta}}||f_{\text{ref}}\right] \right],$$

where the clip function bounds the ratio to stabilize. The KL divergence penalizes deviation from the reference policy and is estimated as $\mathbb{D}_{\text{KL}}\left[f_{\boldsymbol{\theta}}||f_{\text{ref}}\right] = \frac{f_{\text{ref}}(o_{i,t}|q,o_{i,<t})}{f_{\boldsymbol{\theta}}(o_{i,t}|q,o_{i,<t})} - \log\frac{f_{\text{ref}}(o_{i,t}|q,o_{i,<t})}{f_{\boldsymbol{\theta}}(o_{i,t}|q,o_{i,<t})} - 1$. In addition, the ratio $\rho_{i,t} = \frac{f_{\boldsymbol{\theta}}(o_{i,t}|q,o_{i,<t})}{f_{\text{old}}(o_{i,t}|q,o_{i,<t})}$, and the advantage $\hat{A}_i = \frac{r_i - \text{mean}(\{r_i\}_{i=1}^{G})}{\text{std}(\{r_i\}_{i=1}^{G})}$ with the reward:

$$r_i = \lambda_{\text{acc}}R_{\text{acc}}(o_i) + \lambda_{\text{fmt}}R_{\text{fmt}}(o_i) + \lambda_{\text{perc}}R_{\text{perc}}(o_i) + \lambda_{\text{conf}}R_{\text{conf}}(o_i). \tag{5}$$

The key to the above objective lies in the design of the reward function. Our reward function is divided into four parts: (1) the *accuracy* reward is used to guide the model to generate reliable answers; (2) the *format* reward is used to standardize the model's output content; (3) the *perception* reward is used to measure whether the image content perceived by the model (*i.e.*, *<observe>...</observe>*) is correct; (4) The *consistency* reward is used to ensure that the observation content is fully utilized in the reasoning phase and that there is logical consistency between the observation and reasoning content. By balancing these four aspects, the model is encouraged to generate outputs that are not only accurate but also well-structured, context-aware, and logically rigorous.

**Accuracy Reward** $R_{\text{acc}}(\cdot)$ checks whether the model's final answer is correct. A regular expression first extracts the predicted answer from the output, which is then compared to the ground-truth answer. The accuracy reward is binary: the model receives 1 if the prediction matches the ground truth and 0 otherwise. As the evaluation is anchored to the answer segment between *<answer>* and *</answer>*, this reward pushes the model to deliver responses that are both well-formatted and factually accurate.

**Format Reward** $R_{\text{fmt}}(\cdot)$ assesses whether a model's output adheres to a prescribed structure. It is binary: the model earns a reward of 1 only when its response strictly follows the "observe-then-think" pattern of *<observe>... </observe>*, *<think>... </think>*, and *<answer>... </answer>*; otherwise, the reward is 0. By explicitly incentivizing structural compliance, this mechanism enforces regularity in model outputs, keeps the chain of reasoning transparent, and separates the final answer. This reward promotes consistency across tasks and streamlines downstream evaluation and error localization.

**Perception Reward** $R_{\text{perc}}(\cdot)$ measures how closely the model's perception aligns with the ground truth understanding of an image. It first extracts the "observe" content from the model output using a regular expression to capture all text within <observe> tags. The extracted content is then compared to the ground truth using the ROUGE-L metric (Lin, 2004), which evaluates similarity based on the longest common subsequence, balancing precision and recall with scores ranging from 0 to 1.

**Consistency Reward** $R_{\text{conf}}(\cdot)$ evaluates the logical consistency between the <observe> and <think> sections of a model's output, as judged by a reference model, such as Qwen3-4B (Yang et al., 2025). This reward encourages that the reasoning in the <think> section directly builds upon and logically extends the observations made in the <observe> section, improving the consistency of model output.

Table 1: Performance comparison of various models across multiple vision-language datasets. Note that the **boldface** numbers represent the improvements over the base model results. Missing values are denoted by "-". (MME uses sum score)

| Method | Datasets | | | | | | AVG |
| --- | --- | --- | --- | --- | --- | --- | --- |
| | MathVista | WeMath | BLINK | V*Bench | VisuLogic | MME | |
| *Closed Source Models* | | | | | | | |
| GPT-4o | 63.8 | 50.6 | 68.0 | 66.0 | 26.3 | 2329 | 59.6 |
| Claude-3.7-Sonnet | 41.9 | 49.3 | - | - | 24.8 | - | - |
| *Open Source Models* | | | | | | | |
| InternVL2-8B | 58.3 | 26.6 | 50.9 | 63.9 | 25.3 | 2210 | 50.7 |
| MiniCPM-V2.6-8B | 60.6 | 16.4 | 53.0 | 64.9 | 22.9 | 2348 | 50.3 |
| Mulberry-7B | 63.1 | 29.0 | 53.7 | 67.0 | 22.9 | 2396 | 53.5 |
| R1-VL-7B | 63.5 | 29.7 | 52.6 | 51.3 | 24.4 | 2376 | 51.1 |
| Qwen2-VL-7B | 58.2 | 25.6 | 53.2 | 72.8 | 21.9 | 2327 | 52.5 |
| OTT-7B | 65.7 | 33.8 | 53.7 | 72.8 | 24.7 | 2447 | 56.3(**+3.8**) |

## 4 EXPERIMENTS

**Datasets.** Our evaluation datasets include MathVista (Lu et al., 2024), WeMath (Qiao et al., 2024), BLINK (Fu et al., 2024b), V*Bench (Wu & Xie, 2024), VisuLogic (Xu et al., 2025), and MME (Fu et al., 2024a). These benchmarks collectively assess perception and reasoning in VLMs through diverse tasks: BLINK examines core visual tasks like relative depth estimation, visual correspondence, forensics detection, and multi-view reasoning; V*Bench focuses on search in cluttered images; MathVista demands compositional mathematical analysis on visuals; WeMath involves problem decomposition to evaluate knowledge generalization; VisuLogic probes vision-centric logic, avoiding textual biases; and MME evaluates fine-grained comprehension and supporting logical inference. Together, they span modalities like plots, tables, and OCR inputs, along with paradigms such as symbolic, numerical, and spatial reasoning, ensuring a thorough assessment.

**Models.** To evaluate the effectiveness and generality of our proposed method, we conduct experiments on Qwen2-VL-7B-Instruct (Wang et al., 2024), a powerful VLM. Since our evaluation benchmarks span a wide range of visual reasoning tasks, including mathematical problem solving *e.g.*, MathVista, WeMath, core visual perception *e.g.*, BLINK, V*Bench, vision-centric logic *e.g.*, VisuLogic, and general multimodal evaluation *e.g.*, MME, this suite of benchmarks provides a comprehensive validation of our method's effectiveness across varying task complexity and modality diversity.

**Training Setup.** The SFT and RL stages of this study are conducted using four NVIDIA A800 GPUs, each with 80 GB of memory. Both supervised fine-tuning and reinforcement learning stages utilize the same dataset. The supervised fine-tuning stage employs a learning rate of 5.0e-6 and runs for 3 epochs. The reinforcement learning stage is configured with a rollout number of 8, a learning rate of 1.0e-6, and a total of 2 training episodes.

**Evaluation Setup.** To ensure a systematic and reproducible evaluation of VLMs' visual reasoning capabilities, we adopt VLMEvalKit[1] as our evaluation framework. VLMEvalKit offers a unified and extensible interface for benchmarking VLMs across a broad range of datasets and task formats. By building on VLMEvalKit, our results can be comparable and reproducible. The temperature parameter is set to 0.0 to ensure deterministic output generation. For the evaluation of math-related datasets, the powerful Qwen2.5-VL-32B-Instruct model is employed as the judge. The datasets evaluated include MathVista and WeMath, which focus on testing visual reasoning abilities. Other datasets, such as V*Bench, MME, VisuLogic, and BLINK, which have fixed-choice answer formats, do not require additional LLM judges for evaluating the answers.

### 4.1 MAIN EXPERIMENTS

As shown in Table 1, our OTT framework effectively enhances the reasoning and perceptual capabilities of VLMs, tackling the core challenge of comprehensive multimodal reasoning, with

---

[1] https://github.com/open-compass/VLMEvalKit

Table 2: Evaluation results on MathVista and MME with and without the Observe.

| Method | MathVista | MME |
|---|---|---|
| Qwen2-VL-7B | 58.2 | 2327 |
| + GRPO | 59.0 | 2297 |
| + GRPO with Observe | 61.1 | 2385 |

Table 3: Comparison of Base model(Qwen2-VL-7B) with GRPO, SFT, and OTT method.

| Method | V*Bench | VisuLogic |
|---|---|---|
| Base model + GRPO | 71.7 | 24.5 |
| Base model + SFT | 69.1 | 24.5 |
| OTT-7B (Ours) | 72.8 | 24.7 |

Table 4: Ablation Study on Reward Components: Impact of Perception and Consistency Modules on WeMath and MathVista Datasets.

| Perception | Consistency | WeMath | MathVista |
|---|---|---|---|
| ✓ | × | 32.6 | 63.2 |
| × | ✓ | 30.3 | 63.5 |
| ✓ | ✓ | 33.8 | 65.7 |

Qwen2-VL-7B serving as the baseline across six benchmark datasets: MathVista, WeMath, BLINK, V*Bench, VisuLogic, and MME. The baseline model often struggles to integrate perception and reasoning effectively, resulting in suboptimal performance on tasks that demand fine-grained visual understanding and abstract inference. In contrast, our proposed OTT-7B model, leveraging the OTT framework, demonstrates consistent improvements over Qwen2-VL-7B, achieving a 7.5% gain on MathVista and an 8.2% gain on WeMath, alongside an increase of 120 points on MME.

These gains are primarily driven by the SFT stage's structured observe-then-think workflow, and the Perception-Guided Consistency Optimization (PGCO) stage's perception-guided reinforcement learning, which enhances both perceptual accuracy and reasoning depth. On MathVista, for instance, the structured workflow strengthens multi-step mathematical reasoning, while on WeMath, the joint optimization of perception and reasoning improves knowledge generalization in decomposed problems. Additionally, OTT-7B outperforms other models on specific tasks, surpassing Mulberry-7B and R1-VL-7B on MathVista by 2.6% and 2.2%, and on V*Bench by 5.8% and 21.5%, respectively.

Compared to closed-source models, OTT-7B exhibits competitive performance. For instance, on MathVista, OTT-7B outperforms GPT-4o by 1.9% and Claude-3.7-Sonnet by 23.8%; on V*Bench, it exceeds GPT-4o by 6.8%; and on MME, it surpasses GPT-4o by 118 points. However, gaps remain in perception-intensive tasks such as BLINK.

## 4.2 ABLATION STUDIES

**The Importance of Observation in VLMs Reasoning.** Table 2 presents the ablation results of the Observe strategy. Compared with the base model (Qwen2-VL-7B) and the original Vanilla GRPO, incorporating Observe into the GRPO training framework consistently improves performance on both MathVista and MME. This demonstrates that Observe effectively enhances the reasoning capability of VLMs. The Chat Templates used for Vanilla GRPO and Vanilla GRPO with Observe are provided in Appendix F.

**Comparison with GRPO, SFT, and OTT.** Table 3 presents a comparison of the base model (Qwen2-VL-7B) under different training strategies, including Vanilla GRPO, SFT, and our proposed OTT method. While Vanilla GRPO improves performance to 71.7% on V*Bench and 24.5% on VisuLogic, and SFT achieves 69.1% on V*Bench and 24.5% on VisuLogic, OTT-7B delivers superior results with 72.8% on V*Bench and 24.7% on VisuLogic. Validate the effectiveness of our approach in enhancing multimodal reasoning and perceptual accuracy.

**Synergistic Effects of Perception and Consistency Rewards in OTT.** The Table 4 presents a study on the impact of perception reward and consistency reward within the OTT framework, evaluated on WeMath and MathVista datasets. When only the perception reward is applied, performance reaches 32.6% on WeMath and 63.2% on MathVista, underscoring its essential role in enhancing visual processing. Activating only the consistency reward yields 30.3% on WeMath and 63.5% on MathVista, highlighting its contribution to reasoning alignment. The optimal results, 33.8% on WeMath and 65.7% on MathVista, are achieved when both rewards are combined, demonstrating

Table 5: Comparison of OTT's Multimodal Reasoning: 2B vs 7B with 20k SFT Data.

| Method | BLINK | MathVista |
|---|---|---|
| **SFT:20k RL:20k** | | |
| Qwen2-VL-2B | 43.8 | 43.0 |
| OTT-2B | 44.2 (+ 0.4) | 50.2 (+ 7.2) |
| **SFT:20k RL:20k** | | |
| Qwen2-VL-7B | 53.2 | 58.2 |
| OTT-7B | 53.8 (+ 0.6) | 65.7 (+ 7.5) |

Table 6: Comparison of OTT's Multimodal Reasoning: 2B vs 7B with 260k SFT Data.

| Method | BLINK | MathVista |
|---|---|---|
| **SFT:260k RL:20k** | | |
| Qwen2-VL-2B | 43.8 | 43.0 |
| OTT-2B | 44.6 (+ 0.8) | 51.1 (+ 8.1) |
| **SFT:260k RL:20k** | | |
| Qwen2-VL-7B | 53.2 | 58.2 |
| OTT-7B | 53.8 (+ 0.6) | 64.5 (+ 6.3) |

Table 7: Comparison of OTT's multimodal reasoning performance under different SFT data scales.

| Method | BLINK | MME |
|---|---|---|
| **SFT:20k RL:20k** | | |
| Qwen2-VL-7B + SFT | 51.3 | 2344 |
| OTT-7B | 54.7 (+ 3.4) | 2429 (+ 85) |
| **SFT:260k RL:20k** | | |
| Qwen2-VL-7B + SFT | 52.0 | 2339 |
| OTT-7B | 53.8 (+ 1.8) | 2386 (+ 47) |

Table 8: Comparison of OTT's multimodal reasoning performance under different RL data scales.

| Method | BLINK | MME |
|---|---|---|
| **SFT:260k RL:5k** | | |
| Qwen2-VL-7B + SFT | 52.0 | 2339 |
| OTT-7B | 52.4 (+ 0.4) | 2348 (+ 9) |
| **SFT:260k RL:20k** | | |
| Qwen2-VL-7B + SFT | 52.0 | 2339 |
| OTT-7B | 53.8 (+ 1.8) | 2386 (+ 47) |

that integrating perception reward and consistency reward significantly boosts overall multimodal reasoning performance.

**Scalability and Performance Gains of OTT Across Model Scales.** The Table 5 compares the multimodal reasoning performance of the OTT framework across different model scales, using Qwen2-VL-2B and Qwen2-VL-7B as base models. OTT-2B demonstrates a notable improvement over Qwen2-VL-2B, with gains of 0.4% on BLINK and 7.2% on MathVista, highlighting the framework's effectiveness even at smaller scales. Similarly, OTT-7B significantly outperforms Qwen2-VL-7B, achieving increases of 0.6% on BLINK and 7.5% on MathVista, underscoring the scalability of OTT and its ability to enhance reasoning consistency and perceptual accuracy as model capacity grows. The table in Table 6 further examines OTT's performance with an expanded SFT dataset of 260k when $\lambda_{\mathrm{conf}} = 0$, comparing 2B and 7B scales. OTT-2B shows enhancements over Qwen2-VL-2B, with improvements of 0.8% on BLINK and 8.1% on MathVista, indicating robust gains even at the smaller scale with increased training data. Likewise, OTT-7B surpasses Qwen2-VL-7B, achieving gains of 0.6% on BLINK and 6.3% on MathVista, demonstrating that while the larger model benefits from scalability, the incremental improvement diminishes slightly with the larger SFT dataset.

**Impact of Data Scale Variations on OTT Multimodal Reasoning Performance.** Tables 7 and 8 assess the impact of different SFT and RL data scales on the multimodal reasoning performance of the OTT framework when $\lambda_{\mathrm{conf}} = 0$, using Qwen2-VL-7B as the baseline. In Table 7, with SFT data increasing from 20k to 260k and RL data fixed at 20k, OTT-7B improves over Qwen2-VL-7B by 3.4% on BLINK and 85 points on MME with 20k SFT, but the gains decrease to 1.8% and 47 points with 260k SFT, indicating a saturation effect. In Table 8, with RL data scaling from 5k (randomly sampled from the 20k dataset) to 20k and SFT data fixed at 260k, OTT-7B performance enhances, with BLINK improving from 0.4% to 1.8% and MME from 9 to 47 points, highlighting the critical role of RL in optimizing reasoning.

### 4.3 Case Studies

**Case Study Analytical Reasoning in Mathematical Contexts.** This case, Fig. 4, involves a right triangle $\triangle PMN$ with a right angle at $N$ and side lengths $PN = 15$, $NM = 39$, and $PM = 36$. The question requires expressing the ratio of $\tan M$ as a decimal to the nearest hundredth. The initial GRPO analysis attempts to apply the Pythagorean theorem but, due to input errors, computes $\tan M \approx 0.38$. The OTT framework, through structured reasoning steps,

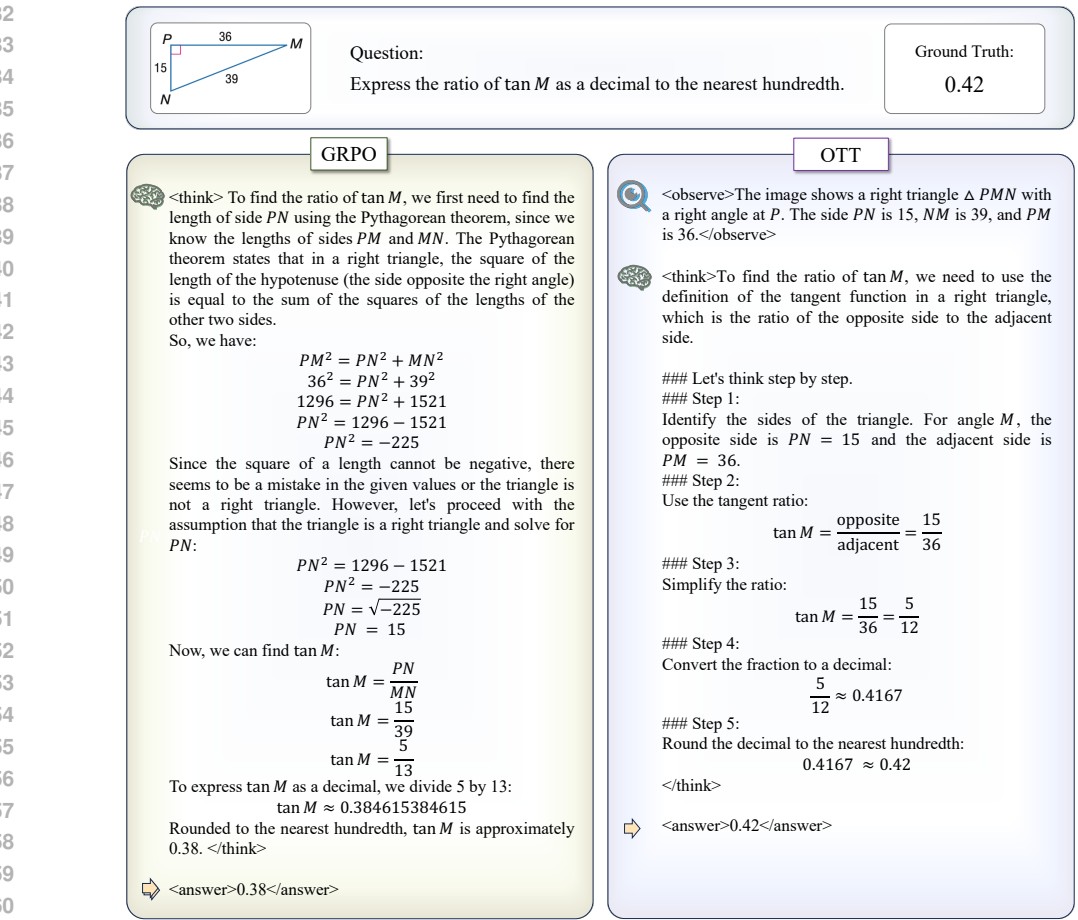

Figure 4: Case study of OTT and GRPO responses to a mathematical reasoning question.

correctly identifies the opposite ($PN = 15$) and adjacent ($PM = 36$) sides for angle $M$, calculating $\tan M = \frac{15}{36} = \frac{5}{12} \approx 0.42$. The final answer is 0.42, aligning with the Ground Truth. This case highlights OTT's ability to correct errors and ensure logical consistency in mathematical reasoning.

## 5 CONCLUSION

**Conclusion.** We proposed a two-stage training framework (OTT) that significantly enhances the performance of vision-language models in complex multimodal reasoning tasks through supervised fine-tuning and perception-guided reinforcement learning optimization. Experimental results demonstrate state-of-the-art performance across six diverse benchmarks, covering tasks involving diagrams, equations, tables, and OCR-heavy inputs. The structured "observe-then-think" not only improves model interpretability but also strengthens robustness in fine-grained multimodal tasks. Our approach highlights the potential of reinforcement learning to jointly optimize perception and reasoning, offering new directions for the advancement of vision-language models. Future work will focus on expanding dataset diversity, refining reward mechanisms, and exploring more efficient training strategies to achieve more comprehensive multimodal reasoning capabilities.

**Limitations.** Although vision-language models (VLMs) have made great strides in combining visual and linguistic data, they still struggle with complex reasoning tasks requiring deep abstract understanding. The main challenge is integrating perception, which handles sensory inputs, with reasoning, which involves symbolic and multi-step inference, and the lack of high-quality multimodal reasoning datasets further limits their real-world performance in areas like scientific analysis.

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

## A    ETHIC STATEMENT

The study does not involve human subjects, data set releases, potentially harmful insights, applications, conflicts of interest, sponsorship, discrimination, bias, fairness concerns, privacy or security issues, legal compliance issues, or research integrity issues.

## B    REPRODUCTION STATEMENT

The experimental setups for training and evaluation are described in detail in Appendix F, and the experiments are all conducted using public datasets. We provide the link to our source codes to ensure the reproducibility of our experimental results: `https://anonymous.4open.science/r/OTT-2DE4`.

## C    IMPACT STATEMENT

Our work advances visual language model (VLM) reasoning through reinforcement learning, enhancing the analysis of complex visuals such as charts and scenes. This approach offers significant potential for scientific data interpretation, robotics, and human-computer interaction. Moreover, it promotes transparency and trustworthiness by generating structured "observe-then-think" outputs. However, it also raises concerns about potential misuse, such as the creation of misleading content, necessitating careful and responsible deployment.

## D    LLM USAGE DISCLOSURE

This submission was prepared with the assistance of LLMs, which were utilized for refining content and checking grammar. The authors assume full responsibility for the entire content of the manuscript, including any potential issues related to plagiarism and factual accuracy. It is confirmed that no LLM is listed as an author.

## E    RELATED WORK

**Supervised Fine-tuning (SFT)** adjusts the policy model to predict the next token based on data that is more closely aligned with the downstream task. The objective of SFT, as shown in Eqn. 6, is to maximize the token-wise log probability of model outputs $o \sim \mathcal{O}(q)$ collected from the training dataset, which are treated as ground truth reasoning paths for questions $q$.

$$\mathcal{J}_{\text{SFT}}(f_{\boldsymbol{\theta}}) \triangleq \mathbb{E}_{(q,a)\sim\mathcal{D}, o\sim\mathcal{O}(q)} \left( \frac{1}{|o|} \sum_{t=1}^{|o|} \log f_{\boldsymbol{\theta}}(o_t|q, o_{<t}) \right). \tag{6}$$

**Proximal Policy Optimization (PPO)** (Schulman et al., 2017) is a widely used actor-critic reinforcement learning algorithm in the fine-tuning stage of large language models. It simplifies the Trust Region Policy Optimization (TRPO) (Schulman et al., 2015a) by maximizing the advantage $A_t$ of model-generated outputs $o$ without requiring ground truth outputs. The advantage $A_t$ is estimated using the generalized advantage estimation (Schulman et al., 2015b), which combines 1) the value of the output as predicted by a learned value model and 2) a KL penalty to regulate the divergence between the current policy model $f_{\boldsymbol{\theta}}$ and the reference model $f_{\text{ref}}$. PPO maximizes the following objective, where $\text{clip}(\cdot, 1-\epsilon, 1+\epsilon)$ ensures that updates do not deviate excessively from the reference policy by bounding the ratio between $1-\epsilon$ and $1+\epsilon$.

$$\mathcal{J}_{\text{PPO}}(f_{\boldsymbol{\theta}}) \triangleq \mathbb{E}_{(q,a)\sim\mathcal{D}, o\sim f_{\text{old}}(\cdot|q)} \frac{1}{|o|} \sum_{t=1}^{|o|} \min\left[ \frac{f_{\boldsymbol{\theta}}(o_t|q, o_{<t})}{f_{\text{old}}(o_t|q, o_{<t})} A_t, \text{clip}\left( \frac{f_{\boldsymbol{\theta}}(o_t|q, o_{<t})}{f_{\text{old}}(o_t|q, o_{<t})}, 1-\epsilon, 1+\epsilon \right) A_t \right]. \tag{7}$$

**Direct Preference Optimization (DPO)** Rafailov et al. (2023) is an approach that directly optimizes the policy model based on human preference data, without requiring the intermediate step of advantage

estimation or the use of a learned value function, as in PPO. Instead, DPO aims to align the model outputs with human preferences by maximizing the likelihood of preferred responses over less preferred ones:

$$\mathcal{J}_{\text{DPO}}(f_{\boldsymbol{\theta}}) \triangleq \mathbb{E}_{(q,o^+,o^-) \sim \mathcal{D}} \left[ \log \frac{f_{\boldsymbol{\theta}}(o^+|q)}{f_{\boldsymbol{\theta}}(o^+|q) + f_{\boldsymbol{\theta}}(o^-|q)} \right]. \tag{8}$$

**Group Relative Policy Optimization (GRPO)** (Shao et al., 2024) simplifies PPO via removing the learnable value model. Instead, GRPO uses the average reward of multiple sampled outputs for the same question. Specifically, given a question $q$, GRPO requires to sample $G$ outputs from the reference model as $\{o_i\}_{i=1}^G \sim f_{\text{old}}(\cdot|q)$. Then, it computes the reward $r_i$ for each output $o_i$ (through deterministic reward functions) and obtains a group of rewards $\{r_i\}_{i=1}^G$. The advantage $\hat{A}_i$ is computed as: $\hat{A}_i = \frac{r_i - \text{mean}(\{r_i\}_{i=1}^G)}{\text{std}(\{r_i\}_{i=1}^G)}$. With $\rho_{i,t} = \frac{f_{\boldsymbol{\theta}}(o_{i,t}|q,o_{i,<t})}{f_{\text{old}}(o_{i,t}|q,o_{i,<t})}$, GRPO maximizes the advantage while ensuring that the model remains close to the reference policy:

$$\mathcal{J}_{\text{GRPO}}(f_{\boldsymbol{\theta}}) \triangleq \mathbb{E}_{(q,a) \sim \mathcal{D}, \{o_i\}_{i=1}^G \sim f_{\text{old}}(\cdot|q)} \frac{1}{G} \sum_{i=1}^G \frac{1}{|o_i|} \sum_{t=1}^{|o_i|} \tag{9}$$

$$\left[ \min \left( \rho_{i,t} \hat{A}_{i,t}, \text{clip} \left( \rho_{i,t}, 1 - \epsilon, 1 + \epsilon \right) \hat{A}_{i,t} \right) - \beta \mathbb{D}_{\text{KL}} \left[ f_{\boldsymbol{\theta}} || f_{\text{ref}} \right] \right].$$

Recent studies have highlighted several limitations of these algorithms. First, SFT, while straightforward, has limited generalization ability due to its reliance on memorizing dataset-collected outputs, and the data collection process can be costly and domain-specific (Chu et al., 2025a). Second, PPO, despite its widespread use, suffers from unstable training, high computational costs, and challenging value model learning due to high variance, making it sample-inefficient and prone to reward hacking (Andrychowicz et al., 2020; Henderson et al., 2018). Third, DPO, while eliminating the need for a separate value model, relies on human preference data, which can be noisy, subjective, and expensive to collect, and its pairwise comparison approach can lead to inconsistencies in preference ranking (Li et al., 2024; Guo et al., 2024). Finally, GRPO, though it simplifies value estimation, can produce incoherent outputs, demands carefully designed reward functions, and remains computationally intensive, sometimes struggling to outperform simpler methods (Guo et al., 2025a; Ouyang et al., 2022).

In addition, variants of DPO (Li et al., 2024; Guo et al., 2024; Xu et al., 2024; Munos et al., 2024; Hong et al., 2024; Xie et al., 2024) rely on human-labeled preference pairs, while approaches like KTO (Ethayarajh et al., 2024) and BCO (Jung et al., 2024) use only single binary labels (e.g., like or dislike). Meanwhile, methods like the Process Reward Model (Uesato et al., 2022; Lightman et al., 2024) and Step-KTO (Lin et al., 2025a) incorporate feedback at each reasoning step for more fine-grained guidance. Recent work further refines the optimization objective of GRPO, including DAPO (Yu et al., 2025), Dr. GRPO (Liu et al., 2025b), REINFOECE++ (Hu, 2025), CPPO (Lin et al., 2025b), and GPG (Chu et al., 2025b).

**RL Post-training for Vision-Language Models.** Recent research on multimodal reasoning converges on two-stage training strategies that blend SFT or curricula with RL to produce structured, logically coherent chains of thought. Specifically, LMM-R1 first performs Foundational Reasoning Enhancement by PPO-optimizing on high-quality text-only CoT data, then enters Multimodal Generalization Training across geometry, science, Sokoban, and football-strategy datasets, achieving large gains on MathVista and M3Exam (Peng et al., 2025b). Curr-ReFT follows a three-phase curriculum—binary, multiple-choice, and open-ended—under GRPO and finishes with rejected-sample tuning, where GPT-4 filters hard, previously failed cases for SFT that boosts linguistic robustness without eroding earlier skills (Deng et al., 2025).

In addition, MM-Eureka demonstrates that RL alone can elicit well-formed reasoning chains from pretrained VLMs, markedly improving math-and-science visual QA without any SFT (Meng et al., 2025). LLaVA-Reasoner mixes SFT on direct-answer and chain-of-thought prompts, then applies DPO to align outputs with preferred reasoning styles (Zhang et al., 2024). Skywork R1V alternates GPT-4o-graded SFT rounds with progressively harder examples and concludes with GRPO to sharpen logical structure and evoke distinct "aha" moments (Peng et al., 2025a). In contrast, GTR begins with outcome-only RL using verifiable rewards, then combats thought collapse via Guided Process Generation: a pretrained VLM cleans messy chains, and the cleaned trajectories are imitated in a

joint PPO- objective (Wei et al., 2025). Across benchmarks that cover geometry, science diagrams, puzzle games, and embodied planning, the six methods report stronger generalization, higher answer accuracy, and noticeably crisper chains of thought, highlighting the synergy between RL reward signals and structured or corrective supervision.

Notably, the reward modeling is pivotal in RL post-training. Rule-based RL founded on GRPO has progressed from text-only reasoning to a broad array of multimodal tasks. DeepSeek-R1 first showed that simple handcrafted rewards—ground-truth consistency and format compliance for objective tasks, plus preference-model scores and linguistic bonuses for subjective tasks—are sufficient to elicit rich reasoning in LLMs without neural reward models (Guo et al., 2025b). Building on this insight, researchers have ported GRPO to vision. In GRPO for Image Captioning, the standard SCST objective is replaced by GRPO, with CIDEr used as the reward metric so that generated captions better match human reference captions (Liang, 2025). Reason-RFT and Visual-RFT introduce a unified reward that combines structural conformance with task-specific accuracy signals across discrete answers, cosine-scored math, function-sequence matching, classification, detection, and grounding (Tan et al., 2025; Liu et al., 2025c).

Embodied-R augments this scheme for video-based spatial reasoning with a logical-consistency term forcing the reasoning path itself to justify the answer (Zhao et al., 2025), while VLM-R1 adapts GRPO to referring-expression comprehension and open-vocabulary detection, adding an odLength penalty that discourages redundant boxes (Shen et al., 2025). Collectively, these studies show that carefully crafted, task-aware rule rewards combined with GRPO can stabilize training, curb reward hacking, and yield interpretable chains of thought across diverse multimodal perception and reasoning challenges with VLMs.

## F    IMPLEMENTATION DETAILS

**Chat Template.** Introduce the structured chat template for the OTT framework, as shown in Fig. 6. The template is designed to guide VLMs in systematically processing multimodal reasoning tasks, ensuring a clear separation of perception and reasoning while maintaining accuracy in the final answer. The chat template explicitly defines the model's response workflow: In contrast, the Fig. 5 represents the conventional template used by other models, featuring a streamlined think-then-answer workflow. First, the model, acting as an "assistant," receives system instructions affirming its supportive role. Then, user input includes visual content (e.g., an image) and a related question. The model's response is organized into three key stages:

- **Observation**: The model first analyzes the visual input, extracting relevant information, and records its observations within `<observe>` tags. This stage ensures accurate perception of the input, laying the foundation for subsequent reasoning.
- **Thinking**: The model then engages in step-by-step reasoning, analyzing the question through an internal monologue and documenting the process within `<think>` tags. This stage emphasizes logical coherence and structured reasoning to avoid errors or contradictions.
- **Answering**: Finally, the model generates a concise final answer, recorded within `<answer>` tags, ensuring clarity and directness in the output.

```
Chat Template

<|im_start|> system
You are a helpful assistant.
<|im_end|>
<|im_start|> user
<|vision_start|><|image_pad|>|vision_end|>{question} You should
FIRST think through the reasoning process step by step as an internal monologue, and
FINALLY provide the final answer. The reasoning process MUST BE enclosed within
<think> </think> tags. The final answer MUST BE enclosed within <answer>
</answer> tags.
<|im_end|>
<|im_start|> assistant
```

Figure 5: Structured Chat Template for a Generic Model, Illustrating the Think-then-Answer Workflow.

---

**Observe Chat Template**

```
<|im_start|> system
```
You are a helpful assistant.
```
<|im_end|>
<|im_start|> user
<|vision_start|><|image_pad|>|vision_end|>
```
{question} You should
FIRST observe the provided image or visual input, THEN think through the reasoning process step by step as an internal monologue, and FINALLY provide the final answer. The observation process MUST BE enclosed within `<observe> </observe>` tags. The reasoning process MUST BE enclosed within `<think> </think>` tags. The final answer MUST BE enclosed within `<answer> </answer>` tags.
```
<|im_end|>
<|im_start|> assistant
```

---

Figure 6: Structured chat template for the OTT model, illustrating the systematic process of visual observation, step-by-step reasoning, and final answer generation.

---

**Prompt Template**

You are a strict judge evaluating the logical consistency between the `<observe>` and `<think>` sections in a multimodal task response. The input will be provided in the format: `<observe>... </observe> <think>...</think>`. Your task is to assign a score of 0, 5, or 10 based on how strongly the `<think>` reasoning relates to and utilizes the details in the `<observe>` section.

Scoring criteria:

**0 points**: Completely unrelated—the `<think>` ignores or contradicts the observed content, or addresses a different topic entirely.

**5 points**: Partially related—the `<think>` references some observed elements but includes extraneous, misaligned, or incomplete logic that doesn't fully build on the observation.

**10 points**: Strongly related—the <think> directly leverages key observed details with precise, coherent steps that logically extend the observation to solve the task. Output only the score as a single number (0, 5, or 10).

---

Figure 7: System Prompt Template for Consistency Reward Scoring.

**Dataset Distribution.** Describe the distribution of the multimodal reasoning datasets used for training and evaluating the OTT framework, as shown in Table 9. The datasets span six categories: Mathematical, Figure Understanding, Math Word Problem, Medical, Science, and Nature World QA, with a total of 260k original records. To accommodate model training needs, we sampled 5k and 20k records for different experimental settings.

**Reward Setting.** Our reward function is defined as $r_i = \lambda_{\text{acc}} R_{\text{acc}}(o_i) + \lambda_{\text{fmt}} R_{\text{fmt}}(o_i) + \lambda_{\text{perc}} R_{\text{perc}}(o_i) + \lambda_{\text{conf}} R_{\text{conf}}(o_i)$, where $\lambda_{\text{acc}} = 0.7, \lambda_{\text{fmt}} = 0.1, \lambda_{\text{perc}} = 0.1$, and $\lambda_{\text{conf}} = 0.1$. We assign the highest weight to the accuracy reward to ensure that the model primarily focuses on task correctness first, which is critical in most real-world applications. Meanwhile, the format and perceptual components receive smaller but non-negligible weights to promote outputs that are not only correct but also syntactically well-formed and perceptually natural. This trade-off enables the model to generate results that are both functionally accurate and user-friendly, thereby improving response quality. The consistency reward utilizes a scoring model with the system prompt outlined in Fig. 7, where scores range from 0 to 10 and are normalized to a 0-1 scale.

## G  CASE STUDIES

**Case Study: Commonsense Reasoning.** This case, drawn from the MME dataset Fig. 8, involves an image of a plate containing strawberries, blueberries, avocados, and bananas. The question asks

Table 9: Dataset distribution across 6 categories with 5k and 20k sampled records.

| Tasks | Datasets | Original | Sampled 5k | Sampled 20k |
|---|---|---|---|---|
| Mathematical | GLLaVA, GEOS, UniGeo, GeoQA Plus, Geometry3K, MathVision, GeoMverse, MathV360K | 55k | 0.8k | 3.4k |
| Figure Understanding | DVQA, DocVQA, FigureQA, PlotQA, ChartQA, InfoVQA, MultiHiertt, LRV-Chart | 116k | 1.8k | 7.2k |
| Math Word Problem | IconQA, TabMWP, CLEVR, CLEVR-Math, Super-CLEVR | 41k | 0.8k | 3.3k |
| Medical | VQA-RAD, PMC-VQA | 2k | 0.2k | 0.7k |
| Science | TQA, AI2D, ScienceQA | 17k | 0.5k | 1.8k |
| Nature World QA | VQA-AS, A-OKVQA, TextVQA, VizWiz, VQA2.0 | 24k | 0.9k | 3.5k |
| Overall | – | 260k | 5k | 20k |

whether someone allergic to bananas can consume the fruits in the image. The initial GRPO analysis erroneously assumes the absence of bananas, yielding an incorrect "Yes" answer. Through the OTT framework's "observe-then-think" workflow, the model first accurately identifies the fruits in the image, confirming the presence of bananas, and then reasons that someone allergic to bananas cannot safely consume the fruits, resulting in a final answer of "No," consistent with the Ground Truth. This case underscores OTT's critical role in enhancing visual perception accuracy.

These case studies demonstrate that the OTT framework, through its "observe-think-answer" work-flow, significantly enhances the perceptual accuracy and reasoning reliability of vision-language models in multimodal reasoning tasks, providing robust support for scientific analysis and complex problem-solving.

972
973
974
975
976
977
978
979
980
981
982
983
984
985
986
987
988
989
990
991
992
993
994
995
996
997
998
999
1000
1001
1002
1003
1004
1005
1006
1007
1008
1009
1010
1011

**Question:**

If I am allergic to banana, can I finish the fruit in the picture? Please answer yes or no.

**Ground Truth:**

No

**GRPO**

<think> The image shows a plate with various fruits, including strawberries, blueberries, and avocado. The question asks if someone who is allergic to bananas can finish the fruit in the picture. Since the plate does not contain any bananas, the person allergic to bananas can safely consume the fruits shown in the picture. </think>

<answer>Yes</answer>

**OTT**

<observe>The image shows a plate with various fruits including sliced avocado, blueberries, strawberries, and sliced bananas.</observe>

<think>The question asks if someone who is allergic to bananas can consume the fruit in the image. The focus is on identifying the presence of bananas and assessing the allergy risk.
### Let's think step by step.
### Step 1:Identify the fruits visible in the image. The image shows bananas, strawberries, avocado, and blueberries.
### Step 2:Determine if the person is allergic to any of these fruits. The question specifically mentions an allergy to bananas.
### Step 3:Conclude whether the person can consume the fruit based on their allergy to bananas.</think>

<answer>No</answer>

Figure 8: Case study of OTT and GRPO responses to an MME commonsense reasoning question.
