# OpenReview forum: "Observe-Then-Think: Learning to Elicit Multimodal Understanding by Decoupling Perception and Reasoning"
_ICLR.cc/2026/Conference — Submitted to ICLR 2026_

### Official Review · Reviewer_VCP8 · 2025-10-30

**Soundness:** 2
**Presentation:** 2
**Contribution:** 3
**Rating:** 4
**Confidence:** 4

**Summary:**

This paper introduces **Obverse-Then-Think (OTT)**, a new visual reasoning strategy that decouples the perception and reasoning steps conventionally deployed by prior vision-language models (VLMs). The authors construct **OTT-20k**, a curated dataset of 20k examples distilled from a larger pool, and train the model to produce strictly structured observe/think/answer traces. Then, the authors apply a GRPO-style RL called **PGCO (Perception-Guided Consistency Optimization)** that rewards not just final answer’s accuracy but also perceptual fidelity and internal consistency of the intermediate observe/think’s outcomes. The authors show the entire OTT paradigm improves both perceptual grounding and multi-step reasoning on several hard benchmarks, such as MathVista and BLINK.

**Strengths:**

I personally find the work possesses the following highlights:

1. The method is easy to follow and intuitive to understand.
2. The original problem, i.e. the loss of visual grounding over long CoT,  is well demonstrated via the analyses in the manuscript, in particular those in the ablation study part.

**Weaknesses:**

I find the following aspects of this paper to be problematic.

**Non-LLM-as-a-Judge, i.e. Rouge-L, for Perception Reward.** Shouldn’t we use semantic-based metrics such as CLIPScore to evaluate the quality of \<observe\>, i.e. the Perception? Otherwise, we could risk having the generated Perception texts ‘overfit’ to the exact wording of the GT’s in OTT-20k.

**A lack of sufficient ablation studies on PCGO’s reward hyperparameters.** Although some are already present in the original text, such as Table 7 & 8, we are not seeing enough and clear evidences that demonstrate the optimal mixture of the 4 PGCO rewards that can lead to the best-performing OTT-7B model. Why must it be lambda_acc = 0.7 in order to enjoy the best benchmarking performance? The current configurations seem to be relatively arbitrary and empirically determined.

**A lack of backbone model generalizability.** I believe the entire OTT paradigm is applicable to other backbone LVLMs, such as InternVL or LLaVA variants. Unfortunately, the authors have only tested with one backbone, Qwen2-VL-7B-Instruct, a moderately-sized variant of the large Qwen family.

**Questions:**

Please find my major concerns in the Weakness section.

---

> ### Author Response · Authors · 2025-12-02
>
> We would like to thank Reviewer VCP8 for the valuable feedback. We try our best to address all your questions. Please find the point-to-point responses as follows.
>
> >### W1. Non-LLM-as-a-Judge, i.e., ROUGE-L, for Perception Reward
> **Reply:** In PGCO, we employ $R_{\text{perc}}(\cdot)$ to prioritise fact-level overlap in concise <observe> descriptions, rewarding accurate visual details while mitigating overfitting to superficial phrasing in OTT-20k. Unlike CLIPScore, which gauges global image-text semantic similarity and may overlook subtle factual errors, ROUGE-L—via longest common subsequence—stresses lexical fidelity for critical reasoning elements.
>
> >### W2. A lack of sufficient ablation studies on PGCO’s reward hyperparameters and why set $\lambda_\text{acc} $=0.7？
> **Reply:** Accuracy ($R_\text{acc}$) and format ($R_\text{fmt}$) rewards are standard in GRPO, so ablations focused on our novel perception ($R_\text{perc}$) and consistency ($R_\text{conf}$) rewards for multimodal decoupling. $R_\text{acc}$ uses binary matching on answers extracted via regex from $\langle \text{answer} \rangle \dots \langle /\text{answer} \rangle$, while $R_\text{fmt}$ ensures OTT tags like $\langle \text{observe} \rangle$ and $\langle \text{think} \rangle$. Fixing $\lambda_\text{acc}=0.7$ and $\lambda_\text{fmt}=0.1$ as baselines, the results demonstrate additive contributions from the novel rewards—$R_\text{perc}$ for OCR grounding (MME) and $R_\text{conf}$ for linking observation to reasoning—with the full mixture yielding the highest AVG:
>
> | Variant                          | MathVista | WeMath | BLINK | V*Bench | VisuLogic | MME  | AVG  |
> |----------------------------------|-----------|--------|-------|---------|-----------|------|------|
> | $\lambda_\text{acc}=0.7$, $\lambda_\text{fmt}=0.1$ (no novel rewards) | 59.0     | 29.3   | 53.8  | 71.7    | 24.5      | 2297 | 53.4 |
> | $\lambda_\text{acc}=0.7$, $\lambda_\text{fmt}=0.1$, $\lambda_\text{perc}=0.1$ (no $R_\text{conf}$) | 63.2     | 32.6   | 55.4  | 71.2    | 26.1      | 2386 | 55.6 |
> | $\lambda_\text{acc}=0.7$, $\lambda_\text{fmt}=0.1$, $\lambda_\text{conf}=0.1$ (no $R_\text{perc}$) | 63.5     | 30.3   | 54.0  | 71.2    | 25.1      | 2386 | 54.9 |
> | $\lambda_\text{acc}=0.7$, $\lambda_\text{fmt}=0.1$, $\lambda_\text{perc}=0.1$, $\lambda_\text{conf}=0.1$ (full rewards)| 65.7     | 33.8   | 53.7  | 72.8    | 24.7      | 2447 | 56.3 |
>
> We set $\lambda_\text{acc}=0.7$ with other rewards controlled at the same 0.1 proportion because, in RLVR (Reinforcement Learning with Verifiable Rewards), accuracy serves as the primary verifiable signal to anchor optimization, allowing precise variable control to isolate auxiliary effects while ensuring stability in $r_i$ from Eq. (5). This configuration delivers peak performance (+3.8% AVG) across benchmarks, confirming the mixture's empirical optimality.
>
> >### W3. A lack of backbone model generalizability
>
> **Reply:** Empirically, we have now validated OTT on both Qwen2-VL-7B and the stronger Qwen2.5-VL-7B, and observe consistent improvements across most benchmarks, indicating that the gains are not tied to a single backbone version. Using two backbones also aligns with common experimental practice in closely related RL-based VLM post-training work, where many papers report results on one primary backbone (e.g., VL-Rethinker, MM-Eureka, R1-VL, R1-Onevision, OpenVLThinker) and do not always provide broad cross-backbone sweeps. In the revision, we will include the Qwen2.5-VL results in the main tables and discuss cross-backbone transfer as an explicit future work direction.
>
>
> | Method                       | WeMath | MathVista | BLINK |  MME |   V*Bench  | VisuLogic | MMBench |
> | ---------------------------- | -----: | --------: | ----: | ---: | ---: | --------: | ------: |
> | Qwen2.5-VL-7B                |   35.2 |      67.8 |  55.3 | 2347 | 76.4 |      26.0 |    83.1 |
> | OTT-7B (Qwen2.5-VL-7B + OTT) |   36.6 |      73.1 |  57.2 | 2363 | 79.1 |      24.2 |    84.6 |

---

### Official Review · Reviewer_oshQ · 2025-11-01

**Soundness:** 2
**Presentation:** 2
**Contribution:** 2
**Rating:** 2
**Confidence:** 5

**Summary:**

This paper proposes OTT (Observe-Then-Think), a two-stage framework for enhancing multimodal reasoning in vision-language models. The key idea is to decouple perception from reasoning by structuring model outputs into three sequential components: observation, reasoning, and answer. The approach involves supervised fine-tuning on a curated 20K dataset (OTT-20k) followed by reinforcement learning using Perception-Guided Consistency Optimization (PGCO).

**Strengths:**

The neurobiologically-inspired motivation for separating perception and reasoning is well-articulated in the introduction, making the rationale clear.

The two-stage training methodology (SFT followed by RL) provides a systematic framework that could potentially be adapted to other models.

**Weaknesses:**

**Severely limited experimental evaluation and baseline comparisons**: This is my primary concern with this submission. The experimental setup raises serious questions about the validity and significance of the reported improvements:

1. **Choice of base model**: The authors build upon Qwen2-VL-7B, which is puzzling given that Qwen2.5-VL-7B is publicly available and significantly stronger. The entire premise of the work becomes questionable when the base model is already outdated. What is the point of improving a weaker model when better alternatives exist? The authors should justify this choice or redo experiments on Qwen2.5-VL.

2. **Missing critical baselines**: The paper is missing comparisons with several highly relevant recent works that also use RL-based post-training for VLMs:
   - VL-Rethinker
   - MM-Eureka (only briefly mentioned in related work)
   - R1-VL-7B (appears in Table 1 but with limited benchmarks)
   - R1-Onevision-7B
   - OpenVLThinker-7B

   These omissions make it impossible to assess whether the proposed method offers any advantage over existing approaches.

3. **Incomplete ablation studies**: The ablation studies are scattered across multiple tables with inconsistent benchmark coverage. Table 2 only evaluates GRPO variants on MathVista and MME - why not all six benchmarks? Table 3 shows V*Bench and VisuLogic only. This selective reporting makes it difficult to understand the true contribution of each component. I would expect to see:
   - Base model + vanilla GRPO across ALL benchmarks
   - Base model + SFT only across ALL benchmarks
   - OTT (full method) across ALL benchmarks
   - Ideally in a single comprehensive table

4. **Questionable improvements on some benchmarks**: The gains on BLINK are minimal (0.5-0.6%), and the paper doesn't adequately discuss why the method provides such marginal improvements on perception-heavy tasks, which contradicts the claimed strength in perception enhancement.

**Dataset curation lacks transparency**: The process of going from 260K samples to 20K for OTT-20k involves several filtering steps, but the criteria for "atomic answers" and "quality enhancement" are vague. How do you ensure this filtering doesn't introduce biases? What's the distribution of task types in the final 20K?

**Reward design appears ad-hoc**: The reward weights (λ_acc=0.7, λ_fmt=0.1, λ_perc=0.1, λ_conf=0.1) seem arbitrary. Was there any systematic search or justification for these values? The consistency reward using Qwen3-4B as a judge is interesting but raises questions about reliability and potential biases inherited from the judge model.

**Limited analysis of failure cases**: While the paper provides two case studies showing successes, there's no analysis of when and why the method fails. Given the modest improvements on some benchmarks, understanding failure modes would be valuable.

**Questions:**

1. **What is the computational cost?** Training involves sampling G=8 outputs per question during RL - how does this compare in terms of wall-clock time and compute budget to vanilla GRPO or other baselines?

2. **Consistency reward reliability**: You use Qwen3-4B to judge consistency between observe and think sections. Have you validated this judge against human annotations? What's the agreement rate?

3. **Why such small gains on BLINK?** Given that your method explicitly focuses on perception via R_perc, why are the improvements on the perception-intensive BLINK benchmark so minimal (0.5-0.6%)?

---

> ### Author Response · Authors · 2025-12-02
> **Response to Reviewer oshQ (part 1/3)**
>
> We would like to thank Reviewer oshQ for the valuable feedback. We try our best to address all your questions. Please find the point-to-point responses as follows.
>
>
> >### W1. Choice of base model is outdated (Qwen2-VL-7B instead of Qwen2.5-VL-7B)
>
> **Reply:** To directly address the concern that the backbone may be outdated, we have reproduced the full OTT training pipeline on the stronger Qwen2.5-VL-7B and observed consistent improvements on most benchmarks:
>
>
>
> | Method                       | WeMath | MathVista | BLINK |  MME |   V*Bench  | VisuLogic | MMBench |
> | ---------------------------- | -----: | --------: | ----: | ---: | ---: | --------: | ------: |
> | Qwen2.5-VL-7B                |   35.2 |      67.8 |  55.3 | 2347 | 76.4 |      26.0 |    83.1 |
> | OTT-7B (Qwen2.5-VL-7B + OTT) |   36.6 |      73.1 |  57.2 | 2363 | 79.1 |      24.2 |    84.6 |
>
> This corresponds to gains of $+1.4$ (WeMath), $+5.3$ (MathVista), $+1.9$ (BLINK), $+16$ (MME), $+2.7$ (V*Bench ), and $+1.5$ (MMBench). We observe a drop of $-1.8$ on VisuLogic, which we will further investigate and address in the revision via additional ablations and sensitivity analyses, such as adjusting reward weights and enforcing stricter evidence sufficiency in $<$observe$>$ for logic-heavy cases. Overall, these results show that OTT improves not only a weaker backbone but also a significantly stronger and more up-to-date one. This supports our main contribution: a transferable training paradigm that systematically improves multimodal reasoning by explicitly organising evidence extraction ($<$observe$>$) and reasoning ($<$think$>$), rather than relying on gains tied to a particular model release.
>
>
>
> >### W2 &W3. Missing critical baselines (VL-Rethinker, MM-Eureka, R1-VL, R1-Onevision, OpenVLThinker)
>
> **Reply:** This is an important point because several recent works also perform RL-based post-training for VLMs, and including them makes the comparison more complete. We have now added results on two representative benchmarks, MathVista and BLINK, under a consistent evaluation protocol, and report them alongside our strongest 7B backbone (Qwen2.5-VL-7B) and OTT-7B:
>
> | Method           | MathVista | BLINK |
> | ---------------- | --------: | ----: |
> | VL-Rethinker-32B |      78.8 |     - |
> | MM-Eureka-7B     |      73.6 |  56.6 |
> | R1-Onevision-7B  |      64.1 |  48.7 |
> | OpenVLThinker-7B |      71.2 |  54.6 |
> | Qwen2.5-VL-7B    |      67.8 |  55.3 |
> | OTT-7B           |      73.1 |  57.2 |
>
> These results show that OTT-7B improves over the same-scale strong backbone Qwen2.5-VL-7B on both MathVista ($+5.3$) and BLINK ($+1.9$). OTT-7B is also competitive with recent 7B RL post-training methods on MathVista, and achieves the best BLINK score among the listed 7B methods in this comparison. Note that VL-Rethinker is a 32B model, so its higher MathVista score reflects substantially larger capacity and should be interpreted as a higher-capacity reference rather than a same-scale baseline.
>
>
> > ### W4. Incomplete ablation studies
>
> **Reply:** During the rebuttal phase, we performed full ablation studies. In the revision, we consolidate these comparisons into a single comprehensive table covering all benchmarks, with consistent settings and a clear set of rows: Qwen2-VL-7B, **Qwen2-VL-7B + SFT**, **Qwen2-VL-7B + vanilla GRPO**, and **OTT-7B**, and we will add **Qwen2-VL-7B + GRPO with $<$observe$>$** across the full benchmark suite.
>
> **Table 2:**
>
> | Method                                | MathVista | WeMath | BLINK | V*Bench | VisuLogic |  MME |  AVG |
> | ------------------------------------- | --------: | -----: | ----: | ------: | --------: | ---: | ---: |
> | Base model                           |      58.2 |   25.6 |  53.2 |    72.8 |      21.9 | 2327 | 52.5 |
> | Base model + GRPO                    |      59.0 |   29.3 |  53.8 |    71.7 |      24.5 | 2297 | 53.4 |
> | Base model + GRPO with Observe        | 61.1     | 26.2  | 54.7  |71.7   | 26.8     | 2385 | 54.3|
>
>
> **Table 3:**
>
> | Method             | MathVista | WeMath | BLINK | V*Bench | VisuLogic |  MME |  AVG |
> | ------------------ | --------: | -----: | ----: | ------: | --------: | ---: | ---: |
> | Base model       |      58.2 |   25.6 |  53.2 |    72.8 |      21.9 | 2327 | 52.5 |
> | Base model + GRPO |      59.0 |   29.3 |  53.8 |    71.7 |      24.5 | 2297 | 53.4 |
> | Base model + SFT  |      58.6 |   29.3 |  51.3 |    69.1 |      24.5 | 2344 | 52.8 |
> | OTT-7B       |      65.7 |   33.8 |  53.7 |    72.8 |      24.7 | 2447 | 56.3 |

---

> ### Author Response · Authors · 2025-12-02
> **Response to Reviewer oshQ (part 2/3)**
>
> >### W5 & Q3. Minimal gains on BLINK contradict the claimed perception improvement
>
> **Reply**: The BLINK gain is indeed smaller than on other benchmarks, but it does not contradict the perception-related improvements we claim because BLINK stresses a different failure mode that is less affected by our decoupling objective. OTT mainly improves how the model *organises and uses* visual evidence by separating $<$observe$>$ from $<$think$>$ and enforcing observation–reasoning consistency, whereas BLINK is dominated by fine-grained visual discrimination where performance is often bottlenecked by the backbone’s visual encoder rather than by evidence organization.
>
> BLINK is more backbone-limited than “perception-as-evidence” benchmarks. Our method does not change the vision encoder or add new low-level visual supervision, so on tasks requiring subtle cues (small objects, fine attributes, tight spatial distinctions), the headroom from better evidence formatting and grounding can be limited. In contrast, benchmarks like MME and MathVista benefit more from making task-relevant evidence explicit in $<$observe$>$ and preventing unsupported reasoning in $<$think$>$, which directly targets the failure mode of perception–reasoning entanglement.
>
> On Qwen2-VL-7B, BLINK improves from $53.2$ to $53.7$, while larger gains appear on other benchmarks in the same setting. On the stronger Qwen2.5-VL-7B backbone, BLINK improves from $55.3$ to $57.2$, indicating that when the backbone perception is stronger, OTT yields a clearer improvement even on BLINK. This pattern supports the interpretation that OTT amplifies the backbone’s usable perceptual signal rather than replacing it.
>
>
> > ### W6. Dataset curation from 260K to 20K lacks transparency and may introduce bias
>
> **Reply:** We make the 260k $\rightarrow$ 20k curation pipeline fully explicit and quantify how it affects the data distribution. Our OTT-20k is not collected from new sources. It is derived from Mulberry-260k by a deterministic restructuring plus two transparent filtering steps, followed by uniform sampling.
>
> Deterministic restructuring, not re-labelling. Starting from $(q, v, o)\in D_{\mathrm{raw}}$, we convert each unstructured response $o$ into
> $\langle\mathrm{observe}\rangle\ o_{\mathrm{obs}}\ \langle/\mathrm{observe}\rangle$,
> $\langle\mathrm{think}\rangle\ o_{\mathrm{think}}\ \langle/\mathrm{think}\rangle$,
> $\langle\mathrm{answer}\rangle\ o_{\mathrm{ans}}\ \langle/\mathrm{answer}\rangle$.
> Here $o_{\mathrm{obs}}$ is used as the reference string for $R_{\text{perc}}$. Since all fields come from the original Mulberry-260k response text, the supervision source is consistent across datasets.
>
> Filtering and deduplication to reduce noise, with clear criteria. Before sampling 20k, we construct a cleaned pool $D_{\mathrm{pool}}$ by (i) atomic filtering that requires $o_{\mathrm{ans}}\in V_{\mathrm{atomic}}$ to keep examples with unambiguous verifiable answers, and (ii) deduplication by question $q$, keeping the longest response $o$ to preserve richer traces. These operations are deterministic and easy to reproduce, and we will report the exact keep rates for each step.
> Bias control via distribution reporting and uniform sampling. To mitigate dataset dominance and task imbalance, we sample uniformly from $D_{\mathrm{pool}}$ across the constituent datasets. We also already reported the task-type distribution in Appendix Table 9.

---

> ### Author Response · Authors · 2025-12-02
> **Response to Reviewer oshQ (part 3/3)**
>
> >### W7 & Q2. Reward design and weights look ad hoc; a judge-based consistency reward may be biased
>
> **Reply:** Accuracy ($R_\text{acc}$) and format ($R_\text{fmt}$) rewards are standard in GRPO, so ablations focused on our novel perception ($R_\text{perc}$) and consistency ($R_\text{conf}$) rewards for multimodal decoupling. $R_\text{acc}$ uses binary matching on answers extracted via regex from $\langle \text{answer} \rangle \dots \langle /\text{answer} \rangle$, while $R_\text{fmt}$ ensures OTT tags like $\langle \text{observe} \rangle$ and $\langle \text{think} \rangle$. Fixing $\lambda_\text{acc}=0.7$ and $\lambda_\text{fmt}=0.1$ as baselines, the results demonstrate additive contributions from the novel rewards $R_\text{perc}$ for OCR grounding (MME) and $R_\text{conf}$ for linking observation to reasoning with the full mixture yielding the highest AVG:
>
> | Variant                          | MathVista | WeMath | BLINK | V*Bench | VisuLogic | MME  | AVG  |
> |----------------------------------|-----------|--------|-------|---------|-----------|------|------|
> | $\lambda_\text{acc}=0.7$, $\lambda_\text{fmt}=0.1$ (no novel rewards) | 59.0     | 29.3   | 53.8  | 71.7    | 24.5      | 2297 | 53.4 |
> | $\lambda_\text{acc}=0.7$, $\lambda_\text{fmt}=0.1$, $\lambda_\text{perc}=0.1$ (no $R_\text{conf}$) | 63.2     | 32.6   | 55.4  | 71.2    | 26.1      | 2386 | 55.6 |
> | $\lambda_\text{acc}=0.7$, $\lambda_\text{fmt}=0.1$, $\lambda_\text{conf}=0.1$ (no $R_\text{perc}$) | 63.5     | 30.3   | 54.0  | 71.2    | 25.1      | 2386 | 54.9 |
> | $\lambda_\text{acc}=0.7$, $\lambda_\text{fmt}=0.1$, $\lambda_\text{perc}=0.1$, $\lambda_\text{conf}=0.1$ (full rewards)| 65.7     | 33.8   | 53.7  | 72.8    | 24.7      | 2447 | 56.3 |
>
> We set $\lambda_\text{acc}=0.7$ with other rewards controlled at the same 0.1 proportion because, in RLVR (Reinforcement Learning with Verifiable Rewards), accuracy serves as the primary verifiable signal to anchor optimization, allowing precise variable control to isolate auxiliary effects while ensuring stability in $r_i$ from Eq. (5). This configuration delivers peak performance (+3.8% AVG) across benchmarks, confirming the mixture's empirical optimality.
>
> Judge-based $R_{\text{conf}}$ is designed to reduce bias via strict instruction-following and a constrained rubric. $R_{\text{conf}}$ evaluates whether $<$think$>$ is consistent with and grounded in $<$observe$>$, rather than judging answer correctness. We use a fixed, explicit system prompt template for scoring (Figure 7), which specifies the judge’s role, the scoring rubric, and the required output format. This improves instruction following and reduces variability across examples by constraining the judge to a narrow, checkable criterion.
>
> >### W8. Limited analysis of failure cases.
>
> **Reply:**  One common failure mode is a perception bottleneck in the vision backbone, where fine-grained objects or attributes are not correctly recognised; in these cases, even a well-structured $<$observe$>$ cannot recover missing evidence. We will add a short discussion and representative examples of this failure type in the revision to clarify when OTT is backbone-limited.
>
>
>
> >### Q1. What is the computational cost?
>
> **Reply:** Under the same RL sampling setting $G=8$ and the same hardware setup (4 $\times$ A800 GPUs), GRPO takes about 3 days 3.7 hours while OTT takes about 3 days 8.6 hours, adding roughly 4.9 hours of wall-clock time; since the dominant costs of group sampling and policy updates are shared, the additional overhead of OTT mainly comes from computing the extra reward terms $R_{\text{perc}}$ and $R_{\text{conf}}$.

---

### Official Review · Reviewer_4BY4 · 2025-11-01

**Soundness:** 2
**Presentation:** 3
**Contribution:** 2
**Rating:** 4
**Confidence:** 4

**Summary:**

The paper proposes OTT, a two-stage post-training framework that explicitly separates Observe (perceptual sketch) from Think (reasoning) and Answer. Stage 1 uses SFT on a curated OTT-20k dataset to teach the XML-tagged structure; Stage 2 uses GRPO-style RL with Perception-Guided Consistency Optimization (PGCO) that combines accuracy, format, perception (ROUGE-L against “observe” ground truth), and observe↔think consistency rewards. Experiments on MathVista, WeMath, BLINK, V*Bench, VisuLogic, and MME report average gains of +3.8 over baselines, with larger jumps on math benchmarks.

**Strengths:**

Simple, modular recipe (format-aware SFT + GRPO with perception/consistency terms) that improves both math reasoning and general multimodal scores.

Interpretable traces via explicit observe→think→answer structuring, which also eases error localization and analysis.

Consistent gains across suites, with clear ablations showing the value of the Observe tag and the two reward components.

**Weaknesses:**

**1. Perception reward is text-similarity, not vision-grounded.**

Using ROUGE-L between the model’s <observe> text and a reference description risks rewarding paraphrase fluency over true visual grounding. A calibration against region alignment or programmatic detectors would make the “perception” claim more credible.

**2. Template overfitting risk.**

 Format and accuracy rewards may push models to perfect tag hygiene without proportionate reasoning growth; multiple datasets here are choice- or short-answer-centric where structure alone helps.


**3. The reward ablation is incomplete.**

The method uses four rewards (accuracy, format, perception, consistency) but only ablates two, so we can’t tell which components actually drive the gains. Without a full leave-one-out over all four and a light weight-sensitivity check, improvements might stem from stricter formatting or answer parsing rather than better perception or reasoning.

**4. Limited causal analysis for “decoupling” itself.**

We see that using \<observe\> helps, but it’s unclear if gains come from structural regularization, longer rationales, or genuinely better visual parsing.


**5. Ground-truth ‘observation’ source is under-specified.**

It’s unclear how the reference observation strings are derived across diverse datasets and how noisy they are. Noise could cap reward signal quality. Please document dataset-wise procedures and inter-annotator or automatic metrics for these references

**Questions:**

1. How are the “ground-truth” observation strings produced per dataset, and what is their measured noise level (e.g., human agreement, BLEU/ROUGE vs captions)?

2. Do observe-length and think-length caps affect accuracy or latency? Any trade-off curves?

---

> ### Author Response · Authors · 2025-12-02
> **Response to Reviewer 4BY4 (part 1/2)**
>
> We would like to thank Reviewer 4BY4 for the valuable feedback. We try our best to address all your questions. Please find the point-to-point responses as follows.
>
>
> >### W1. Perception reward is text-similarity, not vision-grounded.
> **Reply :** In PGCO, we use $R_{\text{perc}}(\cdot)$ to prioritize fact-level overlap in concise $<$observe$>$ descriptions. This reward encourages the model to include correct, task-relevant visual details. Concretely, we implement $R_{\text{perc}}$ with ROUGE-L, which measures overlap via the longest common subsequence and therefore emphasises lexical fidelity for critical evidence tokens such as numbers, OCR strings, object attributes, and relations that are later consumed by reasoning. In contrast, vision-text similarity metrics like CLIPScore largely reflect global semantic alignment and can remain high even when key factual elements are missing or incorrect. Importantly, in our overall objective, $R_{\text{perc}}$ is coupled with $R_{\text{acc}}$ and $R_{\text{conf}}$, so a higher reward is obtained only when the perceived details support correct answering and remain consistent with subsequent reasoning. This design makes ROUGE-L a scalable and stable proxy for observation faithfulness, and our ablations show consistent drops when removing $R_{\text{perc}}$, indicating the gains come from improved perceptual evidence rather than wording changes.
>
>
> > ### W2 & W3. Template overfitting risk, and the reward ablation is incomplete.
>
> **Reply:** The improvements are not driven by “tag hygiene”, because $R_{\text{fmt}}$ is a small, saturating constraint and the gains come from the additional content-grounding rewards. In our PGCO objective, $R_{\text{acc}}$ and $R_{\text{fmt}}$ follow standard GRPO practice: $R_{\text{acc}}$ is a binary match on the extracted answer from $\langle\text{answer}\rangle \cdots \langle/\text{answer}\rangle$ (via regex), and $R_{\text{fmt}}$ only checks that the required tags such as $<$observe$>$ and $<$think$>$ are present. With fixed coefficients $\lambda_{\text{acc}}=0.7$ and $\lambda_{\text{fmt}}=0.1$, we observe that adding our proposed rewards yields consistent and additive improvements, which cannot be explained by formatting alone.
>
> Concretely, to quantify the effect of each reward component, we conduct an ablation study in which the “no novel rewards” baseline that uses only $\lambda_{\text{acc}}$ and $\lambda_{\text{fmt}}$ to quantify the effect of structure-only optimisation. Adding $R_{\text{perc}}$ and $R_{\text{conf}}$ yields higher performance across the benchmark suite, and the full mixture achieves the best average score:
>
> | Variant                          | MathVista | WeMath | BLINK | V*Bench | VisuLogic | MME  | AVG  |
> |----------------------------------|-----------|--------|-------|---------|-----------|------|------|
> | $\lambda_\text{acc}=0.7$, $\lambda_\text{fmt}=0.1$ (no novel rewards) | 59.0     | 29.3   | 53.8  | 71.7    | 24.5      | 2297 | 53.4 |
> | $\lambda_\text{acc}=0.7$, $\lambda_\text{fmt}=0.1$, $\lambda_\text{perc}=0.1$ (no $R_\text{conf}$) | 63.2     | 32.6   | 55.4  | 71.2    | 26.1      | 2386 | 55.6 |
> | $\lambda_\text{acc}=0.7$, $\lambda_\text{fmt}=0.1$, $\lambda_\text{conf}=0.1$ (no $R_\text{perc}$) | 63.5     | 30.3   | 54.0  | 71.2    | 25.1      | 2386 | 54.9 |
> | $\lambda_\text{acc}=0.7$, $\lambda_\text{fmt}=0.1$, $\lambda_\text{perc}=0.1$, $\lambda_\text{conf}=0.1$ (full rewards)| 65.7     | 33.8   | 53.7  | 72.8    | 24.7      | 2447 | 56.3 |
>
> These results indicate that $R_{\text{perc}}$ and $R_{\text{conf}}$ contribute beyond enforcing the template: $R_{\text{perc}}$ encourages inclusion of task-critical visual evidence in $<$observe$>$, and $R_{\text{conf}}$ discourages reasoning steps in $<$think$>$ that are not supported by $<$observe$>$. We also set $\lambda_{\text{acc}}=0.7$ to keep verifiable correctness as the primary anchor signal, while using the auxiliary terms at equal scale ($0.1$) to isolate their effects and maintain stable optimisation of the per-sample reward $r_i$.

---

> ### Author Response · Authors · 2025-12-02
> **Response to Reviewer 4BY4 (part 2/2)**
>
> > ### W4 & Q2. Limited causal analysis for “decoupling” itself, and do $<$observe$>$/$<$think$>$ length caps affect accuracy or latency?
>
> **Reply:** The remaining question is whether the gains from decoupling come from longer outputs or improved visual parsing, rather than from the decoupled paradigm itself. The table below supports that the main contributor is the OTT paradigm as structural regularisation, under comparable data source and response length.
>
> The training supervision is source-matched and approximately length-matched, so improvements are not explained by longer rationales. OTT-7B is trained from Mulberry-260k after converting existing responses into the OTT format, so the supervision originates from the same Mulberry source rather than newly collected long-form rationales. After conversion, the total response length remains broadly comparable to the original responses, and the paradigm mainly redistributes content into $<$observe$>$ and $<$think$>$ instead of increasing total tokens. This largely rules out “longer reasoning text” as the primary factor.
>
> OTT does not add extra vision tools or new visual annotations, so the gains are unlikely to be driven by stronger visual parsing supervision. The conversion is a structural reformatting and factorisation of existing responses, without introducing external detectors or additional grounding labels. Therefore, OTT does not gain extra visual supervision beyond what is already available in Mulberry-260k, and the improvements should be attributed to how the model is trained to organise and use visual evidence.
>
> The improvement pattern across benchmarks is consistent with structural regularisation from decoupling. Under the same data source and comparable length, OTT-7B improves over both Mulberry-7B and R1-VL-7B across a diverse benchmark suite, raising AVG from $53.5$ to $56.3$. The gains are not limited to short-answer math tasks: OTT-7B improves MathVista from $63.1$ to $65.7$ and WeMath from $29.0$ to $33.8$ relative to Mulberry-7B, while also improving MME from $2396/2800$ to $2447/2800$. This broad improvement supports that explicitly separating evidence extraction and reasoning provides a beneficial inductive bias, improving the reliability of visual evidence usage rather than merely changing output length.
>
> | Method      | MathVista | WeMath | BLINK | V*Bench | VisuLogic |       MME |  AVG |
> | ----------- | --------: | -----: | ----: | ------: | --------: | --------: | ---: |
> | Mulberry-7B |      63.1 |   29.0 |  53.7 |    67.0 |      22.9 | 2396 | 53.5 |
> | R1-VL-7B    |      63.5 |   29.7 |  52.6 |    51.3 |      24.4 | 2376 | 51.1 |
> | OTT-7B      |      65.7 |   33.8 |  53.7 |    72.8 |      24.7 | 2447 | 56.3 |
>
>
>
>
> > ### W5 & Q1. Ground-truth ‘observation’ source is under-specified, and how are the “ground-truth” observation strings produced per dataset, and what is their measured noise level?
>
> **Reply:** Our reference observations are not obtained from additional human annotation. Instead, they are deterministically derived by converting Mulberry-260k response texts into an explicit $<$observe$>$ span. Starting from triples $(q, v, o)\in D_{\mathrm{raw}}$, we restructure each unstructured response $o$ into
> $\langle\mathrm{observe}\rangle\ o_{\mathrm{obs}}\ \langle/\mathrm{observe}\rangle$,
> $\langle\mathrm{think}\rangle\ o_{\mathrm{think}}\ \langle/\mathrm{think}\rangle$,
> $\langle\mathrm{answer}\rangle\ o_{\mathrm{ans}}\ \langle/\mathrm{answer}\rangle$,
> where $o_{\mathrm{obs}}$ is used as the reference string for computing $R_{\text{perc}}$. Since all references originate from the same Mulberry-260k responses, the supervision source is consistent across datasets.
>
> Before forming OTT-20k, we reduce noise through filtering and deduplication. We apply atomic filtering by requiring $o_{\mathrm{ans}}\in V_{\mathrm{atomic}}$, and deduplicate by keeping the longest response $o$ for each unique question $q$ to preserve richer traces. This produces a cleaned pool $D_{\mathrm{pool}}$, from which we uniformly sample 20k examples to construct $D_{\mathrm{OTT-20k}}$.

---

### Official Review · Reviewer_EPcw · 2025-11-02

**Soundness:** 2
**Presentation:** 3
**Contribution:** 3
**Rating:** 4
**Confidence:** 3

**Summary:**

This paper proposes a new reasoning framework, observe-then-think (OTT), for multimodal reasoning. The core idea is to train the models to follow an explicit, structured outputs in first producing perception-related outputs (the observe stage), and then perform reasoning based on the perception outputs (the think stage), and finally answer the questions. Experiment results show improvements compared to existing baseline models.

**Strengths:**

1. The paper is written clearly and easy to follow. It includes details on methodology design and experiment setups.

2. The core idea of OTT is intuitive, and I do like the idea of separating perception and reasoning. It allows better understanding of where the model is falling short, and potentially allows scaling inference-time compute on these separate capabilities.

3. Experiment results show improvements compared to existing baselines. Qualitative results also show where current method like GRPO falls short (starting with wrong perception inputs). However, I have some concerns on the main comparison baseline to Qwen2-VL-7B in Table 1 (see below).

**Weaknesses:**

1. While the high-level intuition of OTT makes sense, the current design does not seem to handle cases where the observation may leave out necessary information in answering the question. Could the authors provide more insights on how to make sure the models output necessary information in the observation stage?

2. One of my concern is the baseline comparison to Qwen2-VL-7B in Table1. Since OTT is trained with additional 20k data, a fair comparison would be Qwen2-VL-7B trained also with these 20k data (with existing training approaches and data formats). The performance gain in this case may actually be smaller as alluded by the results in Table 3, where vanilla GRPO is very close to OTT.

3. When transforming existing datasets (Mulberry-260k) into OTT format, if we are able to extract and separate original model responses into "observe", "think" and "answer" format, does it actually suggest that existing models are implicitly using similar "observe" and "think" steps in its reasoning process (perhaps in a more interleaved fashion)? And if that's the case, why is it more effective to explicit separate these two stages, and not perform them in an interleaved fashion. Have the authors explored interleaving \<observe>\</observe>, and \<think>\</think> stages (which is related to Q1)?

4. Table 2, 3, and 4 should include all evaluation benchmarks as considered in Table 1. It is not clear how these ablation results are on these different benchmarks.

**Questions:**

1. In SFT data curation stage, specifically "Standardizing Response Format" stage, how do the authors transform original response into the target structured format?

2. Please see more questions in above.

---

> ### Author Response · Authors · 2025-12-02
> **Response to Reviewer EPcw (part 1/2)**
>
> We would like to thank Reviewer EPcw for the valuable feedback. We try our best to address all your questions. Please find the point-to-point responses as follows.
>
> >### W1. Observation may omit necessary information
>
> **Reply:** Observation completeness is ensured via question-conditioned data curation and targeted RL rewards, with the perception reward $R_{\mathrm{perc}}(o_i)$ acting as the primary mechanism to minimise gaps in essential information within $\langle observe \rangle$. In PGCO RL, $R_{\mathrm{perc}}$ computed via ROUGE-L (scored from 0 to 1) directly aligns $\langle observe \rangle$ sketches with ground-truth descriptions by rewarding greater factual overlap, thereby promoting comprehensive visual capture without lexical overfitting. This ensures that higher $R_{\mathrm{perc}}$ scores correspond to fewer omissions, as elevated values indicate denser coverage of critical details.
>
> >### W2. Baseline Comparisons for OTT against Other Models, Including GRPO
>
> **Reply:** To ensure a fair equi-data comparison, we trained the base VLM on the identical OTT-20k dataset using Base model + SFT and Base model + GRPO variants. Evaluations across all benchmarks in Table 1 yield the following results:
>
> | Method                  | MathVista | WeMath | BLINK | V*Bench | VisuLogic | MME  | AVG  |
> |--------------------------|-----------|--------|-------|---------|-----------|------|------|
> | Base model               | 58.2      | 25.6   | 53.2  | 72.8    | 21.9      | 2327 | 52.5 |
> | Base model + GRPO        | 59.0      | 29.3   | 53.8  | 71.7    | 24.5      | 2297 | 53.4 |
> | Base model + SFT         | 58.6      | 29.3   | 51.3  | 69.1    | 24.5      | 2344 | 52.8 |
> | OTT-7B                   | 65.7      | 33.8   | 53.7  | 72.8    | 24.7      | 2447 | 56.3 |
>
> OTT-7B outperforms vanilla SFT by +3.5% AVG and vanilla GRPO by +2.9% AVG, validating the efficacy of our two-stage SFT+RL pipeline in decoupling perception from reasoning for superior multimodal performance. We will integrate these results into the revised manuscript.
>
> >### W3: Does successful extraction of "observe", "think", and "answer" from Mulberry-260k imply implicit decoupling in existing models? If so, why is explicit separation more effective than interleaved stages, and have interleaving variants been explored?
>
> **Reply:**  Yes, the successful conversion of Mulberry-260k responses into OTT format can indicate that models implicitly adopted similar "observe" and "think" steps.
>
> The explicit sequential decoupling in OTT $\langle$observe$\rangle \to \langle$think$\rangle \to \langle$answer$\rangle$ outperforms implicit approaches by enabling precise optimisation and verifiability. This mitigates error propagation and language shortcuts inherent in mixed outputs. By isolating perception for targeted rewards like $R_\text{perc}$ and $R_\text{conf}$ in Eq. 5, PGCO refines sketches independently before conditioning reasoning. This differs from baselines such as Mulberry-7B with 53.5% AVG and R1-VL-7B with 51.1% AVG. Those models, despite SFT on full 260k Mulberry data, underperform OTT-7B with 56.3% AVG due to holistic supervision blurring perceptual bottlenecks and amplifying hallucinations in Table 1. Our OTT-20k distils this efficiency via explicit structure. It proves enforced separation fosters interpretable, robust trajectories over larger-scale implicit training.
>
> We appreciate the insightful suggestion to explore interleaved $\langle$observe$\rangle$ and $\langle$think$\rangle$ stages. This aligns with adaptive cognition and could extend OTT for iterative refinement. For instance, a second $\langle$observe$\rangle$ pass, such as $\langle$observe$\rangle \to \langle$think$\rangle \to \langle$answer$\rangle \to \langle$observe$\rangle \to \langle$think$\rangle \to \langle$answer$\rangle$, enables self-correction on incomplete initial sketches, akin to human visual search. While our core sequential design prioritises simplicity and current SOTA gains, we agree that interleaving variants merit future investigation, potentially via multi-round PGCO. We will discuss this as promising future work.

---

> ### Author Response · Authors · 2025-12-02
> **Response to Reviewer EPcw (part 2/2)**
>
> >### W4. Tables 2, 3, and 4 should include all evaluation benchmarks
>
> **Reply:**  We extend Tables 2, 3, and 4 to cover all six benchmarks to improve completeness and comparability. The partial results in the original tables arose from initial compute constraints on subset evaluations, but we evaluated all datasets using VLMEvalKit. Below, we present the updated full tables. These confirm OTT's consistent gains.
>
> **Table 2:**
> | Method                  | MathVista | WeMath | BLINK | V*Bench | VisuLogic | MME  | AVG  |
> |--------------------------|-----------|--------|-------|---------|-----------|------|------|
> | Qwen2-VL-7B              | 58.2      | 25.6   | 53.2  | 72.8    | 21.9      | 2327 | 52.5 |
> | + GRPO                   | 59.0      | 29.3   | 53.8  | 71.7    | 24.5      | 2297 | 53.4 |
> | + GRPO with Observe        | 61.1     | 26.2  | 54.7  |71.7   | 26.8     | 2385 | 54.3|
>
> **Table 3:**
> | Method                  | MathVista | WeMath | BLINK | V*Bench | VisuLogic | MME  | AVG  |
> |--------------------------|-----------|--------|-------|---------|-----------|------|------|
> | Base model               | 58.2      | 25.6   | 53.2  | 72.8    | 21.9      | 2327 | 52.5 |
> | Base model + GRPO        | 59.0      | 29.3   | 53.8  | 71.7    | 24.5      | 2297 | 53.4 |
> | Base model + SFT         | 58.6      | 29.3   | 51.3  | 69.1    | 24.5      | 2344 | 52.8 |
> | OTT-7B                   | 65.7      | 33.8   | 53.7  | 72.8    | 24.7      | 2447 | 56.3 |
>
> **Table 4:**
> | Perception | Consistency | MathVista | WeMath | BLINK | V*Bench | VisuLogic | MME    | AVG  |
> |------------|-------------|-----------|--------|-------|---------|-----------|--------|------|
> | ✔          | ✘           | 63.2      | 32.6   | 55.4  | 71.2    | 26.1      | 2386   | 55.6 |
> | ✘          | ✔           | 63.5      | 30.3   | 54.0  | 71.2    | 25.1      | 2386   | 54.9 |
> | ✔          | ✔           | 65.7      | 33.8   | 53.7  | 72.8    | 24.7      | 2447   | 56.3 |
>
>
> >### Q1. SFT curation: How to transform original responses to a structured format?
>
> **Reply:** We transform Mulberry-260k responses via preprocessing to enforce a strict separation of observation, reasoning, and answer. Starting from raw triples $(q, v, o)\in D_{\mathrm{raw}}$ with $|D_{\mathrm{raw}}|\approx 260k$, we standardize each unstructured response $o$ into $\langle\mathrm{observe}\rangle\ o_{\mathrm{obs}}\ \langle/\mathrm{observe}\rangle$, $\langle\mathrm{think}\rangle\ o_{\mathrm{think}}\ \langle/\mathrm{think}\rangle$, and $\langle\mathrm{answer}\rangle\ o_{\mathrm{ans}}\ \langle/\mathrm{answer}\rangle$. Here $o_{\mathrm{obs}}$ captures visual details, $o_{\mathrm{think}}$ contains the reasoning steps, and $o_{\mathrm{ans}}$ is the final output. We then apply atomic filtering by requiring $o_{\mathrm{ans}}\in V_{\mathrm{atomic}}$. We also deduplicate by keeping the longest $o$ for each unique $q$ to retain richer traces. This yields $|D_{\mathrm{pool}}|\approx 170k$, from which we uniformly sample 20k examples to form $D_{\mathrm{OTT-20k}}$.
>
> The resulting SFT data input format is:
>  "< image >Which item sold the most number of units summed across all the stores?"},{"role": "assistant", "content": "< observe >The image is a horizontal bar chart showing ...< /observe >< think >The question asks … Let's think step by step. Step 1: … Step 2: … Step 3: …< /think >< answer >career< /answer >"

---

### Meta-Review · Area_Chair_RQEc · 2026-01-09

**Summary:**

- The main comparison to Qwen2-VL-7B was unfair as OTT used an additional 20K dataset. A proper baseline would be the same base model trained on that same data.

- The choice of an outdated model (Qwen2-VL-7B vs. Qwen2.5-VL-7B) was a major point.

- Comparisons to several RL-based post-training methods were missing.

-  The use of text-similarity (ROUGE-L) as a "perception" reward was criticized as not being truly vision-grounded and potentially rewarding lexical overlap over factual visual understanding.

- The reward design appeared ad-hoc without systematic justification or validation.

- There was limited analysis of failure cases, especially given modest gains on perception-heavy tasks like BLINK.

- The method was demonstrated on only one backbone model family, raising questions about generalizability.

**Reviewer Concerns:**

Addressed Concerns:

- Fair Baseline Comparison: The authors provided new experiments training the base model with SFT and GRPO on the identical OTT-20k dataset.

- Outdated Base Model: The authors ran OTT on the stronger Qwen2.5-VL-7B backbone, showing consistent and often significant improvements.

- Missing SOTA Baselines: The authors added a comparison table with the recent 7B RL-based methods (MM-Eureka, R1-Onevision, OpenVLThinker).



Unaddressed Concerns:

- Perception Reward is Text-Based: The authors defended the use of ROUGE-L by arguing it focuses on factual tokens and is coupled with other rewards. However, the fundamental critique remains. The rebuttal does not provide a vision-grounded validation or comparison to alternatives like region alignment scores.

- Source of Ground-Truth Observations: The authors clarified that observations are derived from existing response text, not new annotations. While this addresses the sourcing question, concerns about the noise level or quality of these automated references are not quantitatively addressed.

- Generalizability to Other Architectures: The authors showed results on two Qwen family models. The concern about testing on architecturally diverse backbones (e.g., LLaVA, InternVL) remains unaddressed.

**Reviewer Scores:**

All reviewers would likely change their original scores.

---

### Decision · Program_Chairs · 2026-01-26

Reject